# It depends: Incorporating correlations for joint aleatoric and epistemic uncertainties of high-dimensional output spaces

**Leonhard F. Feiner** [1, 2]                                                                  *leo.feiner@tum.de*

**Manuel Nickel** [1, 2, 3]

**Martin Menten** [1, 4]

**Laurin Lux** [1, 4, 5]

**Rickmer Braren** [6]

**Daniel Rueckert** [1, 4, 7]

**Georgios Kaissis** [1, 7, 8]

**Johannes Paetzold** [1, 5, 9, †]                                                         *jpaetzold@med.cornell.edu*

**Raphael Rehms** [1, 4, †]                                                                  *raphael.rehms@tum.de*

[†] *Equal contribution*

[1] *AI in Medicine and Healthcare, Klinikum rechts der Isar, Technical University of Munich, Germany*
[2] *Institute of Diagnostic and Interventional Radiology, Klinikum rechts der Isar, Technical University of Munich, Germany*
[3] *Department of Diagnostic and Interventional Radiology and Nuclear Medicine, University Medical Center Hamburg-Eppendorf, Germany*
[4] *Munich Center for Machine Learning (MCML), Munich, Germany.*
[5] *Department of Radiology, Weill Cornell Medicine, New York, USA*
[6] *German Cancer Consortium (DKTK), Munich partner site, Heidelberg, Germany*
[7] *Department of Computing, Imperial College London, London, UK*
[8] *Hasso Plattner Institute for Digital Engineering, University of Potsdam, Potsdam, Germany*
[9] *Cornell Tech, Cornell University, New York, USA*

**Reviewed on OpenReview:** *openreview.net/forum?id=zw5EuUnBny*

## Abstract

Uncertainty Quantification (UQ) plays a vital role in enhancing the reliability of deep learning model predictions, especially in scenarios with high-dimensional output spaces. This paper addresses the dual nature of uncertainty — aleatoric and epistemic — focusing on their joint integration in high-dimensional regression tasks. For example, in applications like medical image segmentation or restoration, aleatoric uncertainty captures inherent data noise, while epistemic uncertainty quantifies the model's confidence in unfamiliar conditions. Modeling both jointly enables more reliable predictions by reflecting both unavoidable variability and knowledge gaps, whereas modeling only one limits transparency and robustness. We propose a novel approach that approximates the resulting joint uncertainty using a low-rank plus diagonal covariance structure, capturing essential output correlations while avoiding the computational burdens of full covariance matrices. Unlike prior work, our method explicitly

combines aleatoric and epistemic uncertainties into a unified second-order distribution that supports robust downstream analyses like sampling and log-likelihood evaluation. We further introduce stabilization strategies for efficient training and inference, achieving superior UQ in the tasks of image inpainting, colorization, optical flow, and depth estimation. See Appendix A for notation used throughout.

# 1 Introduction

In high-risk settings such as AI-supported decision-making, UQ is an essential requirement to support a viable level of reliability and trustworthiness. For instance, AI tools in medicine and healthcare may benefit from a sound UQ (Hüllermeier & Waegeman, 2021; Tran et al., 2022; Band et al., 2022; Gruber et al., 2023; Lopez et al., 2023). However, current Bayesian UQ methods often neglect output correlations in high-dimensional settings, limiting reliability in those tasks (Kendall & Gal, 2017). For instance, tasks on images like optical flow or inpainting imply a high dimensional output, often with more than 1000 dimensions that cannot easily be handled with respect to computation. A common strategy in this context is to distinguish between two types of uncertainty: aleatoric and epistemic. Aleatoric uncertainty is modeled as part of the head of a model, often using distributions like Gaussian for regression. It is generally considered to be irreducible by collecting more information, like increasing the size of the dataset, and therefore can be seen as inherent data noise. In contrast, epistemic uncertainty is reducible and a consequence of a lack of knowledge (Murphy, 2022). For instance, utilizing a large amount of data is expected to reduce epistemic uncertainty.[1] Epistemic uncertainty, due to its complexity, is commonly approximated by sampling from a proxy distribution of models (Hüllermeier & Waegeman, 2021).

Combining both in a single model usually results in a so-called second-order distribution (Bengs et al., 2023). On the one hand, it consists of a distribution over model weights capturing epistemic uncertainty. On the other hand, it models a distribution over plausible predictions representing aleatoric uncertainty. Sampling from the model weights and performing a transformation (forward pass) of the input data results in another distribution representing the aleatoric uncertainty. The shape of this second-order distribution limits further analysis, it is difficult to visualize, and it does not admit a closed-form solution of the marginal likelihood of a sample. Therefore, the second-order distribution is typically marginalized and approximated by a single distribution, representing the joint uncertainty.

Traditionally, uncertainties across outputs have been jointly represented without considering correlations between outputs (e.g., pixels), assuming independent factorized univariate Gaussian distributions. However, neglecting such correlations can limit a comprehensive understanding of uncertainty, especially in scenarios where dependencies between model outputs exist — such as in pixel-wise semantic segmentation (Monteiro et al., 2020), optical flow estimation, image inpainting, or graph node regression.

Figure 1 illustrates the increased representational power of full covariance matrices (right) compared to diagonal ones (left). In both cases, samples from the weight space yield multiple predictions containing a mean and (co-)variance. The expected covariance represents the aleatoric component $\Sigma^a$, while the covariance of means contributes the epistemic component $\Sigma^e$. Their sum yields the joint covariance matrix $\Sigma = \Sigma^a + \Sigma^e$.

Incorporating these correlations efficiently, however, remains challenging. The number of pairwise correlations scales quadratically as $\mathcal{O}(S^2)$ with the number of outputs $S$, resulting in extremely large covariance matrices. This makes many standard operations - such as sampling and computing log-likelihoods - computationally infeasible for high-dimensional outputs.

Consequently, many prominent Bayesian methods focus on low-dimensional output spaces, where exact inference is tractable (Williams & Rasmussen, 1996; Rasmussen & Williams, 2006). Extending these methods to high-dimensional outputs, where considering correlations is essential, remains an open research challenge.

**Related Work**  To estimate epistemic uncertainty, various Bayesian frameworks have been developed, including methods like stochastic variational inference (Blundell et al., 2015), Monte Carlo dropout (Gal &

---

[1]Here, we refer to the standard interpretation of aleatoric and epistemic uncertainty. However, this distinction is not always clear and subject to discussion (Hüllermeier & Waegeman, 2021; Gruber et al., 2023).

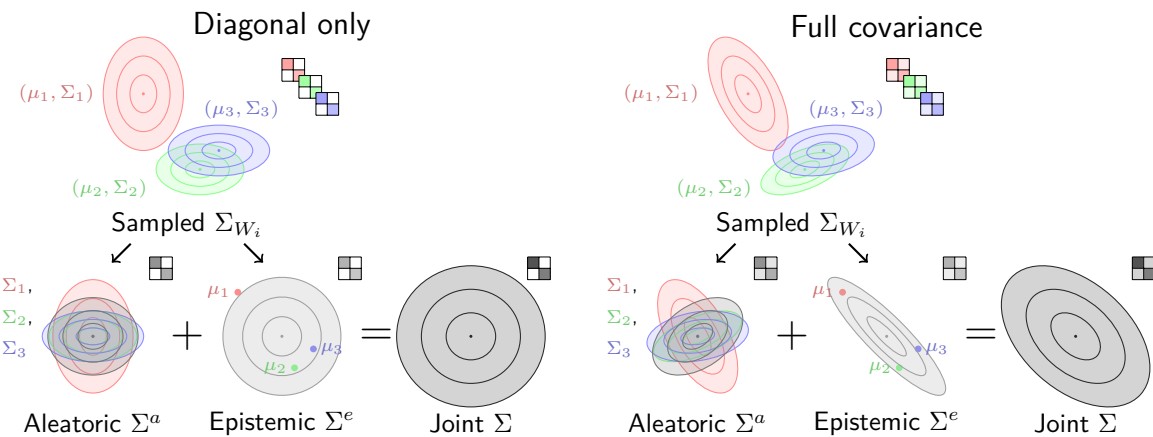

Figure 1: Visualization of covariance matrices for the 2D case. Three predictive samples $(\mu_i, \Sigma_i)$ from a Bayesian model (e.g. sampled from a approximated posterior). The model predicts a mean $\mu_i$ and covariance $\Sigma_i$ for each sample (colors). The columns show the inferred aleatoric $\Sigma^a$, epistemic $\Sigma^e$, and joint $\Sigma = \Sigma^a + \Sigma^e$ uncertainty, respectively. On the left, the covariance matrices are purely diagonal, limiting their representational power. To the right, the same matrices are depicted with non-diagonal values kept, allowing them to capture the overall uncertainty in greater detail.

Ghahramani, 2016), deep ensembles (Lakshminarayanan et al., 2017), stochastic weight averaging (Maddox et al., 2019), or Laplace approximation (Daxberger et al., 2021). The modeling of heteroscedastic aleatoric uncertainty, where the model predicts all parameters of a target distribution and minimizes the corresponding log-likelihood, has also been well established (Nix & Weigend, 1994; Skafte et al., 2019; Stirn & Knowles, 2020; Seitzer et al., 2022). The latter three works further address the challenge of stabilizing training for such networks — a challenge that becomes more critical when modeling structured forms of uncertainty. Building upon these works, others have unified epistemic and aleatoric uncertainty in a single model (Kendall & Gal, 2017; Depeweg et al., 2018; Stirn et al., 2023; Immer et al., 2024; Valdenegro-Toro & Mori, 2022; Mucsányi et al., 2024; Chan et al., 2024; Wimmer et al., 2023). However, all aforementioned methods either evaluate their method only for prediction tasks with a single output value or approximate the marginalized likelihood as a factorized Gaussian, disregarding inter-pixel correlations.

This simplification neglects the inherent dependencies between output dimensions, often resulting in miscalibrated uncertainties and incoherent predictions in structured settings. Modeling these correlations is therefore crucial and has been explored in various applications, including localization (Russell & Reale, 2021), human pose estimation (Gundavarapu et al., 2019), pixel regression (Dorta et al., 2018a;b; Duff et al., 2023), multi class predictions (Willette et al., 2021), and segmentation (Monteiro et al., 2020).

Some approaches that predict full covariance matrices are limited to low-dimensional output spaces (Russell & Reale, 2021; Gundavarapu et al., 2019). To scale to high-dimensional outputs, standard techniques typically sparsify the covariance matrix. However, several such methods are restricted to modeling uncertainty in local neighborhoods via band Cholesky parametrizations (Dorta et al., 2018a;b; Duff et al., 2023). To capture global correlations, recent works (Salinas et al., 2019; Monteiro et al., 2020; Willette et al., 2021; Stoica & Babu, 2023) have successfully employed a low-rank plus diagonal (LR+D) parametrization.

Existing LR+D solutions, most notably Stochastic Segmentation Networks (Monteiro et al., 2020), focus exclusively on modeling aleatoric uncertainty. While other methods learn low-rank factors of aleatoric uncertainty directly without a diagonal component (Nehme et al., 2024; Yair et al., 2024), they result in rank-deficient semi-definite matrices that lack the positive definiteness required for log-likelihood evaluation.

One could argue that models implicitly capture correlations through inherent patterns, similar to latent variable models for aleatoric uncertainty (Depeweg et al., 2018) or deep ensembles for epistemic uncertainty (Lakshminarayanan et al., 2017). However, these methods do not explicitly represent or provide those

correlations. Zepf et al. (2024) are getting close to this goal and combine aleatoric and epistemic uncertainty with a LR+D representation. However, by partially using the Maximum a posteriori (MAP) solution as a further approximation, they do not account for the influence of the model uncertainty on the estimation of the aleatoric uncertainty, ultimately degrading the uncertainty estimate. Furthermore, unlike our approach, they do not resolve the second-order distribution into a unified joint representation. This prevents their method from being used for direct log-likelihood calculation and limits its utility to consecutive sampling.

In conclusion, while significant advancements have been made in modeling covariances for uncertainty estimation, the existing approaches suffer from limitations such as local sparsification, inadequate joint representations, and neglect of epistemic uncertainty, indicating a need for further research to develop more comprehensive and globally accurate uncertainty estimation methods.

**Contribution**   In this work, we propose joint modeling of aleatoric and epistemic uncertainty in a single framework. Unlike existing approaches that approximate the second-order distribution with factorized normals (neglecting output correlations), our method preserves crucial correlations while avoiding the prohibitive space and time costs of full covariance matrices in high-dimensional outputs $\sim 10^4 - 10^7$ outputs). Our low-rank plus diagonal (LR+D) covariance parameterization reduces memory from $\mathcal{O}(S^2)$ to $\mathcal{O}(SR)$ and reduces log-likelihood computation from $\mathcal{O}(S^3)$ to $\mathcal{O}(SR^2 + R^3)$ ($R \ll S$), enabling joint UQ on 65,000-dimensional outputs like CelebA. Furthermore, the low-rank eigenvectors extracted from the covariance provide interpretable insights into dominant modes of correlated uncertainty. We introduce stabilization techniques for robust training and showcase superior performance on high-dimensional tasks such as CelebA colorization (Liu et al., 2015), Flying Chairs optical flow (Dosovitskiy et al., 2015), and NYU Depth V2 (NYU) depth estimation (Silberman et al., 2012).

## 2   Method

We consider supervised learning tasks where we use a neural network $f_w : \mathcal{X} \to \mathcal{Y}$ with an input space $\mathcal{X} \subseteq \mathbb{R}^M$ and a high dimensional output space $\mathcal{Y} \subseteq \mathbb{R}^S$, where $S$ denotes the number of output units, e.g. pixels times the number of output channels. The weights $w \in W \subseteq \mathbb{R}^K$ of the neural network are interpreted probabilistically, meaning we aim to approximate the posterior distribution of the weights $p(w|\mathcal{D}) = p(\mathcal{D}|w)p(w)/p(\mathcal{D})$ allowing to model the epistemic uncertainty after observing a dataset $\mathcal{D} = \{x_i, y_i\}_{i=1}^N$ with $N$ samples. $p(\mathcal{D}|w)$ refers to the likelihood, that represents the aleatoric part of the uncertainty, e.g. by modeling the standard deviation besides a mean value. $p(w)$ is the prior distribution over the weights. Computing $p(w|\mathcal{D})$ is generally not given in closed form and has to be approximated. The most popular methods include Monte Carlo Dropout (MCD), flavors of Stochastic Variational Inference (SVI), and more simple methods like Deep Ensembles (DEs). We highlight, that the proposed method is *agnostic* to the method, as long as we can sample from an approximated posterior distribution, i.e. a proxy distribution $q_\theta^*(w)$ over the weight space $W$, parametrized by $\theta$. To represent the joint uncertainty, for the prediction of unseen output $y$ given new input data $x$, one approximates the posterior predictive distribution $p(y|x, \mathcal{D}) = \int_W p(y|x, w)p(w|\mathcal{D})dw$ using $q_\theta^*$ instead of $p(w|\mathcal{D})$ in combination with Monte Carlo sampling. Specifically, we sample $T$ weights $w_i$ using $w_i \sim q_\theta^*$.

Before addressing the challenges posed by the high-dimensional output space, we provide some intuition for our proposed method by relating it to a standard procedure for estimating empirical covariance. Consider $N$ zero-mean samples organized into a matrix $X \in \mathbb{R}^{M \times N}$, where each column represents a sample and each row represents a feature. The data covariance can be computed straightforwardly via matrix multiplication, i.e., $\widehat{\Sigma} = \frac{1}{N-1}X^\top(X)$. Ignoring the denominator, one can see that as N grows, this multiplication accumulates the total variation across features, which is subsequently averaged. This concept motivates our approach to estimating the full joint uncertainty in a Bayesian Neural Network (BNN). We apply a similar procedure by concatenating the variations stemming from the epistemic and aleatoric components column-wise. By doing so, we implicitly capture the total variation and calculate the full covariance through matrix multiplication. However, since the weight and output spaces are high-dimensional, a direct calculation is computationally infeasible. Therefore, we utilize a LR+D formulation. Section 2.1 provides an overview of the proposed method, while Sections 2.2 and 2.3 detail how the epistemic and aleatoric components are handled.

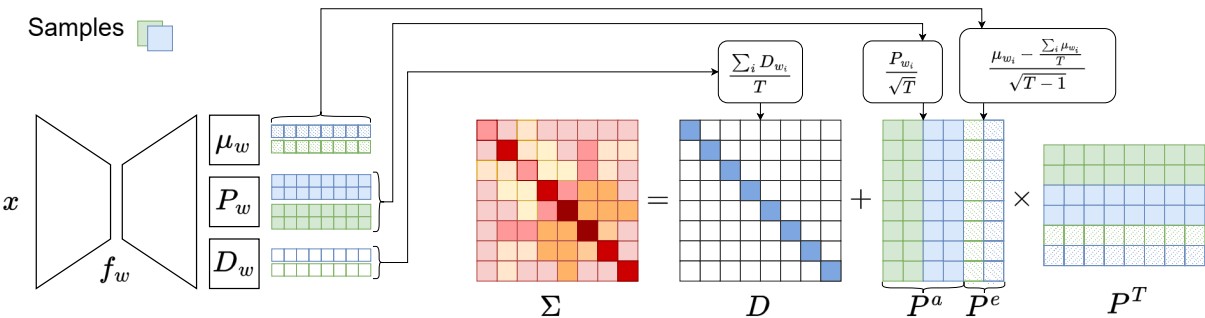

Figure 2: Construction of our LR+D matrix. A network predicts values $\mu_w$, $P_w$, and $D_w$ for two exemplarily sampled weights $w_i$, respectively (green and blue). By averaging the diagonals $D_{w_i}$ and concatenating low rank columns representing the epistemic $P^a$ and aleatoric $P^e$ uncertainty, we build the diagonal $D$ and the low-rank matrix $P$ as parts of our LR+D representation of $\Sigma$. See Section 2 for an in-depth explanation.

## 2.1 Modeling Sparse Joint Uncertainty

To deal with the high dimensionality of $\mathcal{Y}$, we define the likelihood to be a multivariate Gaussian distribution $p(y|x,w) = \mathcal{N}(\mu_W(x), \Sigma_W(x))$, where we keep the spatial complexity of the covariance matrix $\Sigma_W(x)$ low by constructing it in LR+D form. That is, we formulate it as a sum of small matrices, $\Sigma_W(x) = D_W(x) + P_W(x)P_W^\top(x)$, with $D_W$ denoting a diagonal matrix of shape $S \times S$ and $P_W$ a tall matrix of shape $S \times R^W$. We choose a rank $R^W$ much lower than the number of outputs $R^W \ll S$, such that only the most important directions of the aleatoric covariance are covered. We further enforce $D_W$ to contain strictly positive diagonal entries and since $P_W P_W^\top$ is always symmetric, $\Sigma_W$ is always symmetric positive definite by construction and thus a valid covariance matrix. The ultimate goal of this work is to calculate an efficient yet representative representation of the posterior predictive distribution $p(y|x,\mathcal{D})$. We start modeling the parameters of the posterior predictive distribution consisting of mean and covariance by using Monte Carlo integration to approximate the expected model output $\mathbb{E}[y|x,\mathcal{D}] \approx \mu(x)$. The empirical mean is given as $\mu(x) = \frac{1}{T}\sum_i^T \mu_{w_i}(x)$, where $T$ represents the number of weight samples drawn from $w_i \sim q_\theta^\star$.

In our framework, treating model parameters $\theta$ as random variables makes the predicted aleatoric covariance $\Sigma^a$ a random variable as well. This results in a second-order distribution — a distribution over distributions. To arrive at a single, actionable joint representation for downstream tasks, we marginalize out the model parameters $\theta$. By calculating the expectation over the parameter distribution, we resolve this second-order structure into a single predictive covariance matrix $\Sigma$ that accounts for both sources of variation. The joint covariance matrix can be split into epistemic and aleatoric uncertainty using the law of total variance as

$$\underbrace{\text{Cov}[y|x,\mathcal{D}]}_{\substack{\widetilde{\widetilde{\Sigma}}(x) \\ \text{joint uncertainty}}} \approx \underbrace{\text{Cov}_{q_\theta^*}[\mu_W(x)]}_{\substack{\widetilde{\widetilde{\Sigma^e}}(x) \\ \text{epistemic uncertainty}}} + \underbrace{\mathbb{E}_{q_\theta^*}[\Sigma_W(x)]}_{\substack{\widetilde{\widetilde{\Sigma^a}}(x) \\ \text{aleatoric uncertainty}}} . \tag{1}$$

This suggests that the mean of covariance matrices across forward pass samples captures aleatoric uncertainty, whereas the covariance of the means represents epistemic uncertainty. Unlike previous decompositions (Depeweg et al., 2018; Kendall & Gal, 2017) that use variances, our formulation employs covariance matrices, generalizing to multivariate variables. We provide a complete derivation of equation 1 in the Appendix G.4.

Our objective is to represent the joint uncertainty $\Sigma(x)$ in LR+D form as the sum of aleatoric and epistemic uncertainties,

$$D + PP^\top = (D^e + P^e P^{e\top}) + (D^a + P^a P^{a\top}), \tag{2}$$

where $D^a$, $D^e$, and $D$ are diagonal matrices and $P^a$, $P^e$, and $P$ low-rank matrices representing aleatoric, epistemic, and joint uncertainties, respectively. Then, $D = D^e + D^a$ and $P = \begin{bmatrix} P^a & P^e \end{bmatrix}$, where $\begin{bmatrix} & \end{bmatrix}$ denotes

columnwise block concatenation. This expression allows us to conveniently represent both aleatoric and epistemic uncertainties in LR+D form, simplifying further analysis and computation. Figure 2 provides an intuitive illustration about the construction of our LR+D matrix components. Starting with $\Sigma^e$, we describe in detail the individual components of our LR+D representations in the following sections.

## 2.2 Epistemic Uncertainty

The epistemic uncertainty is estimated through the distribution over weights. To derive its covariance, we employ empirical sampling from the proxy distribution over model weights (e.g., SVI (Blundell et al., 2015) or DE (Lakshminarayanan et al., 2017)) as follows:

$$\Sigma^e(x) = \frac{1}{T-1} \sum_i^T \left(\mu_{w_i}(x) - \mu(x)\right)\left(\mu_{w_i}(x) - \mu(x)\right)^\top \quad w_i \sim q_\theta^* \tag{3}$$

Our objective is to avoid the full covariance matrix and instead seek a representation in LR+D form.

To bring the approximated epistemic covariance matrix into LR+D form, we set the diagonal $D^e(x)$ to zero and rewrite the covariance matrix as $\Sigma^e(x) = P^e(x)P^e(x)^\top$, where $P^e(x) \in \mathbb{R}^{S \times R^e}$ has $R^e = T$ columns and is defined as

$$P^e(x) = \frac{1}{\sqrt{T-1}} \left[\mu_{w_1}(x) - \mu(x) \quad ... \quad \mu_{w_T}(x) - \mu(x)\right]. \tag{4}$$

In high-dimensional scenarios, the number of samples is often significantly lower than the number of output dimensions ($T \ll S$), which renders the empirical covariance matrix $\Sigma^e$ low rank and therefore singular. Acquiring a sufficient number of samples to obtain a full-rank empirical estimate is typically infeasible due to time and space complexity constraints. In our low-rank-plus-diagonal parameterization, the epistemic part is captured purely by the low-rank term $P^e$ and the diagonal is zero ($D^e(x) = \mathbf{0}_S$), while the aleatoric diagonal has strictly positive entries ($D_{ii}^a(x) > 0$; cf. Eq. 6). Consequently, the total covariance $\Sigma(x)$ remains positive definite and thus invertible.

## 2.3 Aleatoric Uncertainty

Similar to epistemic uncertainty, the covariance matrix capturing aleatoric uncertainty $\Sigma^a(x)$ can be approximated through empirical sampling. We calculate the empirical mean of covariance matrix estimations over all sampled model weights via

$$\Sigma^a(x) = \frac{1}{T} \sum_i^T \Sigma_{w_i}(x) \quad w_i \sim q_\theta^*. \tag{5}$$

We here again intend to represent $\Sigma^a(x)$ in LR+D form.

To rewrite the covariance matrix containing the aleatoric uncertainty in LR+D representation, we reformulate $\Sigma^a(x) = D^a(x) + P^a(x)P^a(x)^\top$ using

$$D^a(x) = \frac{1}{T} \sum_i^T D_{w_i}(x) \tag{6}$$

$$P^a(x) = \frac{1}{\sqrt{T}} \left[P_{w_1}(x) \quad ... \quad P_{w_T}(x)\right]. \tag{7}$$

This yields a $P^a \in \mathbb{R}^{S \times (T \cdot R^W)}$ with $R^a = T \cdot R^W$ columns. Although $R^a$ generally remains far below $S$, it can still become fairly large as the number of Monte Carlo samples $T$ increases. Thus, we suggest reducing the number of columns of $P(x)$.

## 2.4 Truncated Singular Value Decomposition Approximation

The full matrix $P(x)$, representing the joint uncertainty, uses $R = T \times (R^W + 1) = R^a + R^e$ columns, where each forward pass $i = 1, \ldots, T$ contributes one column from $\mu_{w_i}$ and $R^W$ columns from $P_{w_i}$. In general,

increasing the number of forward passes $T$ yields a better uncertainty representation, as more samples enhance the empirical covariance estimate. However, in this naive representation, larger sample sizes also result in quadratic scaling of computational complexity. Hence, we suggest further approximations to cope with moderately high sample sizes.

Assuming that samples are often correlated and exhibit dominant directions of variance, we propose to reduce the dimensionality of $P(x)$ with truncated Singular Value Decomposition (SVD). Keeping only the most informative columns of $P(x)$ will improve the efficiency of further computations without losing much information. However, the calculation of SVD comes with its own computational complexity that has to be taken into account. Specifically, we decompose the matrix $P$ as $P^\top = U\Psi V^\top$, where $U$ and $V$ are orthogonal matrices, and $\Psi$ is a diagonal matrix containing the singular values in non-decreasing order $\Psi_{1,1} \leq ... \leq \Psi_{S,S}$. Subsequently, we define the matrix $\tilde{P} = V\Psi$ and rewrite the matrix product as $\Sigma = PP^\top = \tilde{P}\tilde{P}^\top$. To reduce dimensionality, we discard the smallest singular values and their associated columns in $V$. However, we keep the univariate variance parts of these dropped columns by transferring them to a new diagonal matrix $\hat{D}$. Hence, the approximated matrix $\hat{\Sigma} = \hat{D} + \hat{P}\hat{P}^\top$ keeps all independent variance and the most important covariances of $\Sigma$. If we keep the $\hat{R}$ largest singular values, the components of $\hat{\Sigma}$ are

$$\hat{P} = \begin{bmatrix} V_{R-\hat{R}} \cdot \Psi_{R-\hat{R},R-\hat{R}} & ... & V_R \cdot \Psi_{R,R} \end{bmatrix} \tag{8}$$

and

$$\hat{D}_{ii} = D_{ii} + \sum_{j=1}^{R-\hat{R}-1} V_{ij}^2 \cdot \Psi_{j,j}^2 . \tag{9}$$

The number of columns $\hat{R}$ to retain is determined by reconstruction error and downstream task performance, as validated in our ablation studies (Sec. 3.3). The aforementioned approach enables us to effectively represent joint uncertainty in the LR+D form. For analysis purposes, SVD can also be applied to both of the low rank summands of $P$ namely the aleatoric part $P^a$ and the epistemic part $P^e$ separately. This allows to visualize the most important directions of variance of both components. See Appendix F for pseudocode.

## 2.5   Stability Considerations

The high dimensionality of the output space $S$ and the inversion of the joint covariance matrix $\Sigma$ present significant numerical challenges. As $S$ increases, the condition number $\kappa(\Sigma)$ tends to grow, potentially leading to numerical non-positive definiteness during the Cholesky decomposition of the capacitance matrix $C = I_R + P^\top D^{-1} P$. Furthermore, as the rank $R$ increases through Monte Carlo sampling, the dimensionality of $C$ grows, and its condition number $\kappa(C)$ tends to increase. To ensure robust training and reliable inversions, we combine active regularization with efficient diagnostic monitoring:

**Condition Number Regularization**   To actively limit the condition number and ensure numerical stability, we regularize the diagonal matrix by setting $D(x) = \epsilon + \exp(Z(x))$. Since the smallest eigenvalue of the joint covariance is bounded below by the smallest entry of the diagonal ($\lambda_1(\Sigma) \geq \min(D)$), the hyperparameter $\epsilon$ serves as a guaranteed theoretical floor. This preventively stabilizes the inversion of the capacitance matrix $C$ by ensuring that $D^{-1}$ remains bounded, effectively capping the ratio between the captured correlations in $P$ and the noise floor.

**Efficient Stability Monitoring**   While $\epsilon$ provides the functional safety margin, we monitor the numerical health of the representation using two metrics. First, we compute the condition number of the capacitance matrix $\kappa(C)$. Since $C$ is of size $R \times R$, this is computationally manageable. Second, for a global assessment of the full covariance $\Sigma$, we utilize Weyl's inequality to evaluate tractable diagnostic bounds for $\kappa(\Sigma) = \lambda_S(\Sigma)/\lambda_1(\Sigma)$:

$$\frac{\lambda_S(PP^\top) + \lambda_1(D)}{\lambda_{R+1}(D)} \leq \kappa(\Sigma) \leq \frac{\lambda_S(PP^\top) + \lambda_S(D)}{\lambda_1(D)}, \tag{10}$$

where $\lambda_S(PP^\top) = \|P\|_2^2$ is the largest eigenvalue of the low-rank component and $\lambda_i(D)$ are the sorted diagonal entries. These bounds provide a real-time diagnostic of the global stability margin with negligible overhead.

**Training Objective**  We further stabilize the optimization using a weighted objective $\mathcal{L} = \mathcal{L}_I + \alpha \mathcal{L}_{\mathrm{lrd}}$ to prevent factor explosion in early training phases:

$$\mathcal{L}_{\mathrm{lrd}} = -\sum_i \log \mathcal{N}\left(y_i \mid \mu(x_i),\, D(x_i) + P(x_i)P^\top(x_i)\right) \tag{11}$$

$$\mathcal{L}_I = -\sum_i \log \mathcal{N}\left(y_i \mid \mu(x_i),\, I\right) \tag{12}$$

The univariate term $\mathcal{L}_I$ acts as an identity-covariance prior, providing a stable gradient signal before the correlations in $P$ are well-defined.

Numerical validation of the Weyl bounds using exact float64 computation on MNIST (where $S$ is small enough for full-rank inversion), along with a sensitivity analysis of how rank $R$ and diagonal floor $\epsilon$ influence training stability, is provided in Appendix E. Furthermore, we provide practical guidelines and robustness techniques in Appendix E.3 to assist practitioners in scaling this framework to high-dimensional tasks.

## 3   Experiments

**Proposed Method**  We empirically evaluate our method of joint aleatoric and epistemic uncertainty modeling using our LR+D representation in several experiments. In all experiments, we use variants of the U-Net (Ronneberger et al., 2015) architecture. We equip the U-Net with probabilistic outputs via three established Bayesian approximation methods: 1) adding dropout, which we use for MCD (Gal & Ghahramani, 2016), 2) DE (Lakshminarayanan et al., 2017), or 3) by using variational convolutional layers for SVI (Blundell et al., 2015), to estimate a distribution over model weights which estimates epistemic uncertainty. However, we note that our approach is compatible with any Bayesian method as long as it is computationally feasible for considered models. We use a combined model to jointly predict mean and uncertainty because it provides a cleaner, more unified architecture with shared feature learning and simpler training, which can — but does not always — lead to better results. Details and comparisons to variants with separated mean and uncertainty estimation can be found in Appendix D.4. Further reproducibility details in Appendix B. The official implementation and scripts to reproduce all experiments are available at `https://github.com/LeonhardFeiner/corr-joint-ae-uq`.

**Datasets and Tasks**  We evaluate our method in different settings on the CelebA, Flying Chairs, and NYU datasets for the tasks of inpainting, colorization, optical flow and depth estimation.

To evaluate our method on facial images, we use the CelebA-HQ dataset, keeping the original splits from celeba (Liu et al., 2015). The original split contains 24,183 images for training, 2,993 for validation, and 2,824 for testing (image size $256 \times 256$). We study two tasks on this dataset: colorization and inpainting.

To evaluate performance on optical flow estimation, we use the Flying Chairs (Dosovitskiy et al., 2015) dataset. This dataset is resized to 192 x 256 and split into 18,297/2,287/2,288 training/validation/test images. We provide visualizations of the predictions as part of the Appendix D.1.

We additionally evaluate on the NYU dataset for the depth estimation task. Following common practice, we resize all images to $240 \times 320$ and use the official test split with 654 samples for evaluation. From the remaining data, we create a self-defined split of 695 training and 174 validation samples.

**Baselines**  We evaluate non-Bayesian models alongside various Bayesian methods. In addition to the diagonal (D) covariance matrix approach from (Kendall & Gal, 2017), we introduce a low-rank plus diagonal (LR+D)-parameterized distribution that captures richer output correlations, representing a substantial advancement in uncertainty modeling.

As a further baseline, we follow the approach by Zepf et al. (2024) and approximate the aleatoric uncertainty term of Equation 1 to prevent sampling aleatoric $P$ matrices. This reduces the number of resulting columns from $T \times (R+1)$ to $T+R$. It is achieved by approximating the expectation of the aleatoric uncertainty $\Sigma^a$ term with the aleatoric covariance prediction of the model with the expected weights:

$$\Sigma^a = \mathbb{E}_{q_\theta^*}\left[\Sigma_W(x)\right] \approx \Sigma_{\mathbb{E}_{q^*}[W]}(x)$$

To compute this term, we require the expected weights of the Bayesian models to be well-defined. For MCD, this is done by turning dropout off and rescaling the activations accordingly. For SVI, where the weights follow Gaussian distributions, the expected weights are simply the means of the Gaussian distributions. For DE, we are unable to define expected weights, hence this approximation is not evaluated in this case. Note that Zepf et al. refer to this approach as MAP solution, which coincides with the *expected weights* solution if the weight uncertainty is modeled with symmetrical unimodal distributions like Gaussians as commonly used by SVI and Laplace Approximation (LA). Furthermore, Zepf et al. do not provide a joint representation, and log-likelihood calculation is only possible using a combination of our methods.

**Model Specifics**  Finally, we evaluate our joint LR+D parametrization in combination with all three Bayesian methods. For this case, we let the model predict a matrix $P_W \in \mathbb{R}^{S \times R^W}$ of rank $R^W = 8$ and for epistemic models, we draw $T = 64$ samples. The predictions are multivariate Normal distributions, represented by their LR+D parametrization. Those predictions are joined to a single, LR+D parametrized distribution. For the full joint uncertainty LR+D model, this yields a joint $P$ matrix with $R = T \times (R^W + 1) = 576$ columns, which we optionally compress down with TSVD while keeping the diagonal variance of the dropped columns as described in Section 2. For the expected weights baseline, we perform an additional forward pass using the expected weights and concatenate the aleatoric and epistemic columns, which leads to $R = 72$ in total. All models are trained for the same amount of steps. The LR+D models are trained with loss weighting of $\alpha = 0.125$ and a minimum diagonal entry of $\epsilon = 0.01$.

**Metrics**  We evaluate our models using metrics that capture both predictive fit and the quality of uncertainty estimates. As our primary quantitative measure, we report test log-likelihood (TLL) of the predictive distribution on held-out data. The TLL is a strictly proper scoring rule for multivariate regression, ensuring that the metric is maximized only when the predicted mean and covariance exactly match the true data-generating distribution (Gneiting & Raftery, 2007). Because the TLL is influenced by both the accuracy of the mean and the calibration of the covariance, we additionally report standard $L_1$ and $L_2$ errors to isolate and assess the point-prediction performance (see App. D.3). These metrics serve as a baseline to ensure that improvements in TLL are not merely a result of variance scaling but reflect a high-fidelity reconstruction. In addition, we consider two complementary uncertainty metrics. First, we use the diagonal of the joint covariance $\Sigma_{ii}$, which contains the per-pixel marginal variances and provides a pixel-wise uncertainty map. Second, we compute the differential entropy of the joint multivariate normal as a scalar measure of overall predictive uncertainty, which we employ for selective prediction via coverage–risk curves (see Sec. D.2).

### 3.1  Main Results - Comparison of the Fit of Predictive Distributions

**Quantitative Results**  We evaluate the TLLs across various dataset–task combinations to assess predictive fidelity. Quantitatively, we find that modeling epistemic uncertainty consistently improves the likelihood of unseen test sample predictions compared to non-Bayesian baselines, as shown in Table 1. This improvement is robust and holds for both the diagonal and LR+D covariance parameterizations across all experiments.

Notably, the expected weights $\mathbb{E}[w]$ approximation provides a consistent performance gain over diagonal-only baselines for all evaluated tasks. However, our proposed method—which explicitly combines both uncertainty types into a unified joint multivariate representation—is superior in every task. While the margin of improvement varies depending on the task's complexity and output dimensionality, our approach consistently outperforms all other tested Bayesian methods and approximations. This demonstrates that capturing the full interaction between aleatoric and epistemic components via a structured covariance is essential for high-fidelity predictive modeling in high dimensions.

**Qualitative Results**  Figure 3 and Appendix Figures 6-9 (Datasets), and 10 (Bayesian Methods) provide qualitative results, where we show how the 8 most important columns in our joint low-rank matrix $P$ describe the areas of correlated uncertainty. For color images, the eigenvectors reveal a structured correlation pattern where pixels with the same directional offset from the mean (both brighter or both darker than neutral gray) exhibit strong positive correlation. Conversely, areas with opposing offsets — where one region is brighter and another is darker — show distinct anticorrelation, reflecting the model's global adjustment of contrast and luminance. To further clarify these relationships in single-channel tasks, such as depth estimation, we employ

| Parameters | Epistemic | Method | CelebA | | Flying Chairs Optical Flow ×1000 | NYU Depth ×1000 |
|---|---|---|---|---|---|---|
| | | | Inpainting ×100 | Colorization ×1000 | | |
| D | ✗ | | -496 ± 39 | 223 ± 26 | -235 ± 11 | -3288 ± 756 |
| LR+D | ✗ | | -600 ± 67 | 455 ± 47 | -188 ± 3 | -1716 ± 75 |
| D | MCD | | -348 ± 20 | 300 ± 30 | -221 ± 5 | -1252 ± 234 |
| | SVI | | -380 ± 35 | 336 ± 5 | -232 ± 6 | -541 ± 237 |
| | DE | | -249 | 374 | -213 | -332 |
| LR+D | MCD | $\mathbb{E}[W]$ | -129 ± 14 | 565 ± 1 | -170 ± 2 | -203 ± 18 |
| | SVI | $\mathbb{E}[W]$ | -207 ± 27 | 565 ± 7 | **-161** ± 5 | -282 ± 49 |
| | MCD | (ours) | -74 ± 13 | 587 ± 2 | -174 ± 1 | -51 ± 10 |
| | SVI | (ours) | -174 ± 29 | 581 ± 4 | **-164** ± 6 | -174 ± 27 |
| | DE | (ours) | **-50** | **589** | -172 | **-13** |

Table 1: Quantitative Results. We evaluate the TLLs (base 10) of model predictions across various dataset–task combinations, standard deviation across 5 model seeds (with the exception of DE, where we report results from a single ensemble due to the high computational cost of training multiple independent ensembles). Higher values correspond to greater likelihood and therefore better predictive performance. Our approach is assessed on four tasks: inpainting, colorization, optical flow, and depth estimation. Likelihoods scale linearly with output dimensionality and, in the case of masking, are computed only over masked regions. Results are reported for both Bayesian (MCD, SVI, DE) and non-Bayesian (✗) networks with diagonal (D) and low-rank plus diagonal (LR+D) covariance parameterizations. For the combination of LR+D and Bayesian methods, we additionally report outcomes using the expected weights $\mathbb{E}[w]$ approximation. Overall, incorporating epistemic uncertainty and the LR+D representation increases TLL, indicating improved predictive fidelity.

a diverging colormap. This replaces the standard grayscale representation with a red-blue scale, allowing for a more intuitive visualization of positive and negative offsets and highlighting how the model distributes uncertainty across different spatial regions. For example, in the CelebA colorization task (top), the visualized $P$ matrix reveals that the first two eigenvectors represent global color shifts across the entire image: the first corresponds to an image wide color axis between orange and blue (complementary colors), and the second to a purple–green axis (also complementary). The third and fourth eigenvectors capture contrast between the foreground (faces) and background. The fifth and sixth focus on variations in hair and eye color.

This demonstrates a hierarchical structure in the learned correlations: the initial, dominant eigenvectors capture global, coarse-scale variations, whereas subsequent factors represent increasingly fine-grained and localized features. The singular values $\Psi$ provide further insight into the relative importance of these correlations, confirming the transition from global to local uncertainty structures. Visualization of the eigenvectors is only possible with our method, which includes the covariance terms; hence, allowing the identification of image regions with correlated uncertainty. The parameter D (Diag) captures additional uncertainty, which could not be captured by the low rank covariance matrix created by $PP^\top$. In summary, these qualitative results can help to intuitively describe the uncertainty relationships at image scale.

## 3.2 Complexity of Covariance Parametrizations

Figure 4 presents the memory (left) and time (right) requirements for computing the log-likelihood of different covariance parameterizations: sparse options like diagonal (D) and low-rank plus diagonal (LR+D), as well as full covariance $\Sigma$ using both naive and lower-triangular parameterizations. The complexity is shown as a function of the number of variables in the covariance matrix, with specific points marking the number of variables for each dataset-task combination. Random numbers were used for all parameters, independent of datasets, for complexity evaluation. For LR+D, we evaluate various numbers of columns $R$ in the low-rank matrix $P$. We limit our analysis to sizes that fit within a single 48GB GPU. As seen in the figure, the LR+D parameterization (with 64 columns) is significantly more efficient than the naive full covariance, both with respect to memory and time, for all datasets. In larger datasets like CelebA and Flying Chairs, the full

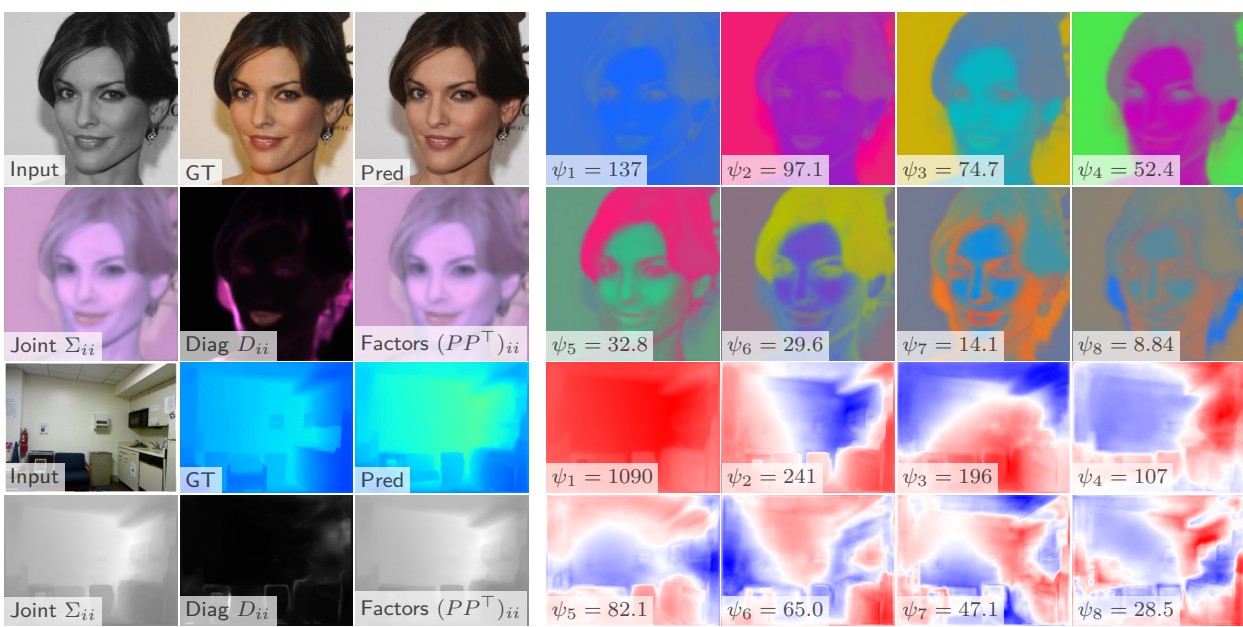

Figure 3: Qualitative Results. Random samples from the test sets depict the input, prediction, ground truth, and parameters of the predictive distribution. The top rows show colorization on CelebA images, while the bottom rows display depth estimation on NYU images. Our model predicts a mean (Pred), the parameter $D$ (Diag), and a low-rank matrix $P$. For each example, the first three columns show, in the top row, the input, ground truth, and prediction, and, in the bottom row, the corresponding marginal variances given by the diagonal of the joint covariance $\Sigma_{ii}$, the diagonal contribution from $D_{ii}$, and the diagonal induced by the low-rank part $(PP^\top)_{ii}$. Columns 4–7 in both rows visualize the first eight leading covariance factors from the joint low-rank matrix $P$, in a random orientation and in descending order of the associated singular values $\Phi$, illustrating dominant directions of correlated uncertainty. We observe that these visualizations highlight regions and structures with elevated uncertainty in both local and globally correlated patterns. For more qualitative results of all datasets and Bayesian methods, please see the Appendix Figures 6-10.

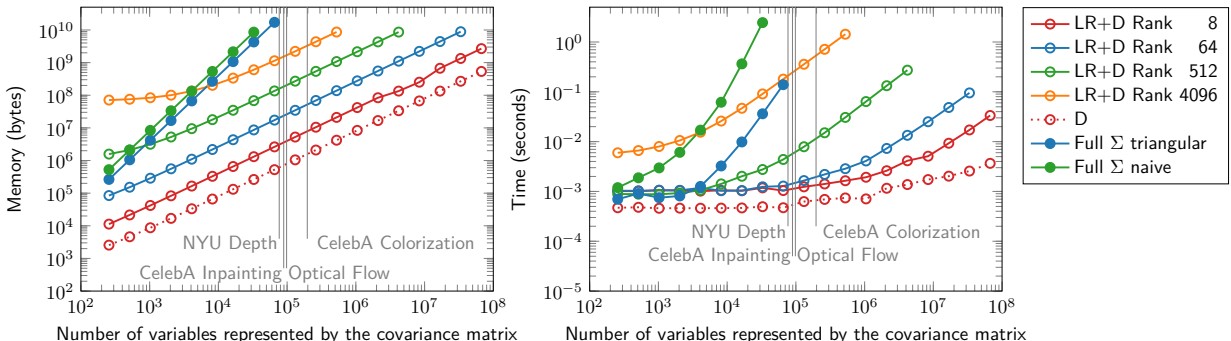

Figure 4: Empirical memory (left) and time (right, avg. of 100 calculations) for log-likelihood across covariance parameterizations. Memory scales linearly for LR+D, quadratically for full $\Sigma$ (until GPU limit). Random parameters used independent of datasets. LR+D handles larger matrices.

covariance matrix approaches the GPU memory limit, even without batching or storing the model and its gradients. Theoretical details on the computational complexity can be found in the Appendix C.

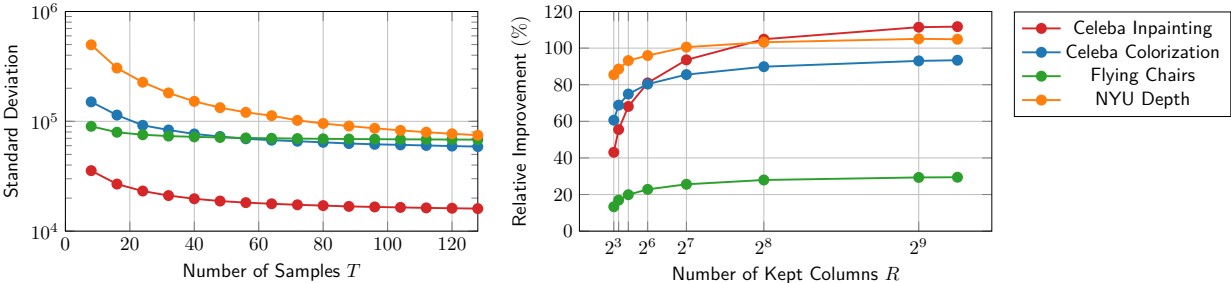

Figure 5: Impact of Sample Count and Low-Rank Truncation on the Stability and Accuracy of LR+D-Parametrized Covariance Approximations. Left: Standard deviation of test log-likelihood decreases with more samples, indicating improved consistency. Right: Relative TLL improvement (%) over diagonal covariance versus retained columns in low-rank approximation.

### 3.3 Effect of Evaluation Hyperparameters

For a comprehensive evaluation of our uncertainty framework, we conduct multiple ablations to identify which factors most influence model performance. Specifically, we study the effects of sample count and dimensionality reduction in the LR+D-parametrized covariance, which sparsely models joint uncertainty.

In figure 5, the left plot shows the average standard deviation of the TLL across the dataset as the number of samples $T$ used to estimate the joint multivariate normal distribution increases. We observe that higher sample counts consistently reduce TLL variability, resulting in more stable and reliable predictions. However, this increases computation for both forward passes and LR+D rank.

In the right plot, we fix the sample count at $T = 64$ and analyze the impact of truncating the number of retained columns in the LR+D covariance structure via TSVD. The y-axis shows the relative improvement in TLL (in %) compared to a purely diagonal covariance. Retaining more columns generally improves performance by capturing a richer covariance structure. Notably, across the board, any compromise involving a limited number of columns still yields significant improvements over a purely diagonal covariance.

We further ablate design choices in Appendix D.4 and compare against Nehme et al. (2024) (Tables 7 and 8). First, as shown in Table 9, incorporating the diagonal update defined in Equation 9 leads to more robust predictions. Next, we investigate the impact of the number of columns $R^W$ in $P_W$, which are directly predicted by the model weights, while keeping the number of columns $R$ retained after TSVD fixed (Table 10). We find that the optimal value of $R^W$ is task-dependent. Nevertheless, for simplicity, stability, and computational efficiency, we adopt a fixed value of $R^W = 8$ across all tasks in our remaining experiments. Furthermore, we provide a comprehensive overview of different design choice combinations in the full ablation Table 11. Finally, we analyze the distribution of eigenvalues of $PP^\top$ and $D$ under different approximations (figure 12).

## 4 Discussion

**Conclusion** In this work, we have explored the dual nature of uncertainties — aleatoric and epistemic — and their integration in high-dimensional regression tasks. We proposed a novel method that employs a low-rank plus diagonal covariance matrix to approximate joint uncertainty, effectively preserving vital output correlations and significantly reducing the computational demands that are inherent to full covariance matrix representation. Our approach lowers memory usage and improves the efficiency of both sampling and log-likelihood calculations. To address stability during training, we incorporate tools to monitor and regularize the condition number of both the covariance matrix and the internally used capacitance matrix.

Empirically, our approach outperforms the commonly used factorized Gaussian representation. It exhibits a higher log-likelihood and produces more reliable uncertainty estimates, demonstrating clear advantages in uncertainty modeling. Beyond quantitative gains, the low-rank structure also exposes interpretable patterns in correlated uncertainties — offering insights into how uncertainty propagates across high-dimensional

outputs. These results highlight the method's effectiveness and interpretability in capturing and quantifying uncertainty in large-scale regression tasks.

**Limitations**   Our method conceptually extends to any Bayesian framework; however, for simplicity and computational reasons, we restrict our evaluation to using Monte Carlo Dropout, Stochastic Variational Inference and Deep Ensemble. Future research into more sophisticated Bayesian inference techniques will likely further enhance the empirical quality of these uncertainty estimates.

Beyond vision, the proposed parameterization is directly applicable to other multivariate regression settings with structured outputs, such as graph-based prediction of node or edge attributes, multivariate time series forecasting of correlated signals (for example, energy demand across regions or multiple physiological channels), and multi-output tabular or scientific regression where several related physical or environmental quantities are predicted jointly. In particular, our method is critical for scenarios requiring well-calibrated uncertainty for downstream spatial aggregation, such as calculating total tracer concentrations or lesion volumes in medical imaging or energy consumption of distributed users in a power grid. In such cases, modeling correlations is essential to prevent uncertainty from being artificially underestimated during the summation process. While directly applicable, we lack empirical evaluation on these domains, which we leave for future work.

The method is flexible with regard to the choice in number of columns utilized in the LR+D-parameterization of the covariance matrix. Increasing the number of columns generally leads to improved uncertainty estimation but comes at the cost of additional conceptual and computational complexity, as well as potential training instability. Training difficulty also arises from numerical sensitivity associated with the covariance matrix and the internally used capacitance matrix. Specifically, both matrices can become ill-conditioned, resulting in numerical errors, particularly during Cholesky factorization. Monitoring and managing the condition number of these matrices is essential to ensure convergence.

While our method introduces computational overhead compared to factorized approaches, this cost is a necessary trade-off for high-dimensional tasks where full covariance matrices are physically intractable. For instance, whereas a $10^5$-dimensional covariance requires $\sim$40GB of memory — surpassing the limits of many modern GPUs —our LR+D parameterization maintains linear scaling. This efficiency ensures the method remains tractable for outputs up to $10^7$ dimensions, making structured uncertainty modeling feasible at scale.

Finally, our method builds upon the assumption that uncertainties in output can be modeled by a single multivariate Gaussian, even though this approximation is often used in the literature (Kendall & Gal, 2017; Monteiro et al., 2020; Duff et al., 2023). However, multivariate Gaussians may not be a suitable approximation for every task, for example, for uncertainties in translation or rotation in images. Exploring epistemic uncertainty under different distributions is a highly promising research question. Furthermore, our Gaussian assumption implies that the distribution is unimodal, where the mean and mode coincide. This may not hold for complex tasks where the ground truth is inherently multimodal. In such cases, the mean estimate may represent an average of several plausible modes rather than a single physically consistent realization.

By more expressive approximation of the posterior predictive distribution than traditional joint distributions, our method enhances both the reliability and explainability of predictions from deep learning models.

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

# A  Symbols and Acronyms

## A.1  List of Symbols

| Symbol | Remark |
|---|---|
| $\mathcal{X}$ | Input space, $\mathcal{X} \subseteq \mathbb{R}^M$ |
| $\mathcal{Y}$ | Output space, $\mathcal{Y} \subseteq \mathbb{R}^S$ |
| $\mathcal{D} = \{(x_i, y_i)\}_{i=1}^N$ | Training dataset with $N$ input–output pairs |
| $\boldsymbol{x} = \{x_i\}_{i=1}^N$ | Stacked input sample from train split $\boldsymbol{x} \in \mathbb{R}^{N \times M}$ |
| $\boldsymbol{y} = \{y_i\}_{i=1}^N$ | Stacked input sample from train split $\boldsymbol{x} \in \mathbb{R}^{N \times S}$ |
| $x$ | Single input (test or train) sample |
| $y$ | Single output (test or train) sample |
| $p(w)$ | Prior distribution over weights |
| $p(\mathcal{D} \mid w)$ | Likelihood of the data given weights |
| $p(w \mid \mathcal{D})$ | Posterior distribution over weights |
| $q_\theta^*(w)$ | Variational (proxy) distribution approximating $p(w \mid \mathcal{D})$ |
| $p(y \mid x, w)$ | Predictive distribution given input $x$ and weights $w$ |
| $p(y \mid x, \mathcal{D})$ | Bayesian predictive distribution (weights marginalized) |
| $p$ | Generic probability distribution |
| $q$ | Generic proxy / variational distribution |
| $N$ | Number of training samples |
| $S$ | Number of output units (e.g., pixels $\times$ channels) |
| $M$ | Number of input units (e.g., pixels $\times$ channels) |
| $T$ | Number of samples drawn from distribution over weights |
| $K$ | Size of weight space |
| $R$ | Number of columns of tall matrix $P$ |
| $\hat{R}$ | Number of columns of $\hat{P}$ after truncation |
| $f_w$ | Neural network mapping $f_w : \mathcal{X} \to \mathcal{Y}$ parameterized by $w$ |
| $W$ | Weight space, $W \in \mathbb{R}^K$ |
| $w$ | Neural network weight vector $w \in W$ |
| $\mu$ | Mean prediction output of the network $x$ |
| $\Sigma$ | Covariance (uncertainty) output of the network |
| $e$ | Prediction error of a sample |
| $D$ | Diagonal matrix used for linear low-rank decomposition (LRD), $D \in \mathbb{R}^{S \times S}$ |
| $\epsilon$ | Minimal variance entry of $D_{ii}$ enforced by implementation |
| $P$ | Tall matrix used for LRD, $P \in \mathbb{R}^{S \times R}$ |
| $C$ | Capacitance matrix used for inversion of $\Sigma$ and log-likelihood calculation, $C \in \mathbb{R}^{R \times R}$ |
| $I_R$ | Identity matrix, $I_R \in \mathbb{R}^{R \times R}$ |
| $\mathbf{0}_S$ | Zero matrix, $\mathbf{0}_S \in \mathbb{R}^{S \times S}$ |
| $U$ | Left singular vectors of $P$, $U \in \mathbb{R}^{R \times R}$ |
| $\Psi$ | Diagonal matrix of singular values of $P$, $\Psi \in \mathbb{R}^{R \times R}$ |
| $\hat{P}$ | Truncated matrix after applying SVD , $\hat{P} \in \mathbb{R}^{S \times \hat{R}}$ |
| $\hat{D}$ | Updated $D$ matrix after applying TSVD to $P$ to keep the variance, $\hat{P} \in \mathbb{R}^{S \times S}$ |
| $\tilde{P}$ | Rotated matrix after applying SVD, $\tilde{P} \in \mathbb{R}^{S \times R}$ |
| $P^\star$ | Orthogonal matrix before normalization (used in ablation), $P^\star \in \mathbb{R}^{S \times R}$ |
| $\bar{P}$ | Orthonormal matrix after normalization (used in ablation), $\bar{P} \in \mathbb{R}^{S \times R}$ |
| $P^W$ | Raw model output matrix (used in ablation), $P^W \in \mathbb{R}^{S \times R}$ |
| $\mathcal{L}$ | Loss function |
| $\mathcal{N}$ | Normal (Gaussian) distribution |
| $\mathbb{E}[\cdot]$ | Expectation operator |
| $\mathrm{Cov}[\cdot]$ | Covariance operator |
| $[.\ .]$ | Column-wise block concatenation |
| $(\cdot)^\top$ | Matrix or vector transpose |
| $(\cdot)^a$ | Aleatoric uncertainty only |
| $(\cdot)^e$ | Epistemic uncertainty only |
| $(\cdot)^W$ | Raw model head output from a single forward pass |
| $(\cdot)_w$ | Quantity viewed as a function of the weights $w$ |
| $(\cdot)_{w_i}$ | Quantity evaluated at a particular weight sample $w_i$ |
| $(\cdot)_{i\_}$ | $i^{\text{th}}$ row of a matrix |
| $(\cdot)_{\_i}$ | $i^{\text{th}}$ column of a matrix |
| $\lambda(\cdot)$ | Eigenvalue operator applied to matrix $\cdot$ |
| $\kappa(\cdot)$ | Condition number operator applied to matrix $\cdot$ |
| $\alpha, \beta$ | Hyperparameters scalar controlling trade-off in loss |
| $\mathcal{L}, \mathcal{L}_I, \mathcal{L}_{\mathrm{lrd}}, \mathcal{L}_{\bar{P}}, \mathcal{L}_{p^a}$ | Loss functions as defined in the equations |
| $\lfloor . \rfloor$ | Stop-gradient operator |

Table 2: List of symbols used in the paper.

### A.2 List of Acronyms

Flying Chairs

**BNN** Bayesian Neural Network

**CelebA** CelebFaces Attributes

**D** diagonal

**DE** Deep Ensemble

**GT** ground truth

**LA** Laplace Approximation

**LL** log-likelihood

**LR+D** low-rank plus diagonal

**MAP** Maximum a posteriori

**MC** Monte Carlo

**MCD** Monte Carlo Dropout

**MNIST** Modified National Institute of Standards and Technology database

**NLL** negative log-likelihood

**NPPC** Neural Posterior Principal Components

**NYU** NYU Depth V2

**RGB** Red, Green, Blue

**SVD** Singular Value Decomposition

**SVI** Stochastic Variational Inference

**TLL** test log-likelihood

**TSVD** Truncated Singular Value Decomposition

**UQ** Uncertainty Quantification

## B Reproducibility

The official implementation and scripts to reproduce all experiments are available at
`https://github.com/LeonhardFeiner/corr-joint-ae-uq`. Checkpoints for all models trained with the proposed method, as well as all baselines, are available upon request. The datasets used in the experiments are publicly accessible and links as well as preprocessing scripts are included in the repository. An extensive schematic, with pseudocode and intuitive description of the method, along with proofs, is also included here. Additionally, qualitative examples are provided to enhance understanding of the method. All experiments were conducted on a single NVIDIA Quadro RTX 8000 GPU with 48 GB of RAM. We use seed 42 for baseline evaluations. To assess stochastic variability and form Deep Ensembles, we extend our training to a set of five seeds $\{42, 43, 44, 45, 46\}$. For consistency across methods, the individual models trained for seed-variability analysis are identical to those utilized as members of the Deep Ensemble.

## C Detailed Computational Analysis and Implementation Considerations

**Computation, Time and Space Complexity** Table 3 gives the theoretical time and memory complexities of various covariance parametrizations and calculations. The sparse representations are more efficient in terms of memory and computational complexity. However, they do not provide all degrees of freedom of a covariance matrix and are limited to either local or the most important global correlations.

**Structural Advantages and Inductive Biases** A significant practical advantage of the LR+D parameterization—unlike full-rank or triangular formulations—is its compatibility with architectures utilizing a homogeneous feature space. In settings where a constant number of channels is predicted for every grid position or graph node, the low-rank component $P \in \mathbb{R}^{S \times R}$ can be implemented efficiently as a position-wise operation, such as $1 \times 1$ convolutions or node-wise linear layers. This implementation allows the model to capture global correlations while preserving the structural inductive biases of the underlying network, a property that is lost when parameterizing a full lower-triangular Cholesky factor.

| Type | Parametrization | Captured correlation | Precompute | Per-Eval $p(y \mid \mathcal{N})$ $x\Sigma^{-1}x$ | $\|\Sigma\|$ | $y \sim \mathcal{N}$ | Memory |
|---|---|---|---|---|---|---|---|
| full Russell & Reale (2021) | correlation | all | $\mathcal{O}(S^3)$ | $\mathcal{O}(S^2)$ | $\mathcal{O}(S)$ | $\mathcal{O}(S^2)$ | $\mathcal{O}(S^2)$ |
| full Gundavarapu et al. (2019) | lower-triangular Cholesky | all | – | $\mathcal{O}(S^2)$ | $\mathcal{O}(S)$ | $\mathcal{O}(S^2)$ | $\mathcal{O}(S^2)$ |
| sparse Dorta et al. (2018a;b) | inverse band Cholesky $\star$ | local | – | $\mathcal{O}(SR)$ | $\mathcal{O}(S)$ | $\mathcal{O}(SR)$ | $\mathcal{O}(SR)$ |
| sparse Monteiro et al. (2020) | LRD | global | $\mathcal{O}(SR^2 + R^3)$ | $\mathcal{O}(SR)$ | $\mathcal{O}(SR)$ | $\mathcal{O}(SR)$ | $\mathcal{O}(SR)$ |
| factorized Kendall & Gal (2017) | diagonal | none | – | $\mathcal{O}(S)$ | $\mathcal{O}(S)$ | $\mathcal{O}(S)$ | $\mathcal{O}(S)$ |

Table 3: This table depicts the computational complexity for calculations using different parametrizations for covariance matrices. Here $S$ is the output dimensionality (number of variables) and $R$ denotes the structural size: for banded Cholesky $R$ is the bandwidth (number of subdiagonals), and for low-rank plus diagonal (LR+D) $R$ is the factor rank; throughout $R \ll S$ is assumed. We use the sparse LR+D parametrization as the basis for our method. This reduces time and spatial complexity in comparison to the naive or Cholesky decomposition and allows for global correlation in comparison to the sparse inverse band Cholesky parametrization. The type and amount of correlations of different parametrization is different (Captured Corr). Furthermore, the used representation enables for efficient calculation of $\Sigma$ or $\Sigma^{-1}$ (Parametr. Representation) and needs different amount of memory. The time complexity is given for calculation of the mahalanobis distance $x\Sigma^{-1}x^{\top}$, determinant $|\Sigma|$ as well as sampling.

$\star$ Inverse band Cholesky assumes a direct parametrization of a banded precision factor $L$ with $\Sigma^{-1} = LL^{\top}$. The per-evaluation costs shown (quadratic form and sampling in $\mathcal{O}(SR)$, log-determinant in $\mathcal{O}(S)$) use precision-based sampling via triangular solves, as in Dorta et al. (2018a;b). Achieving these $\mathcal{O}(SR)$ costs in practice requires implementations that exploit band structure; generic dense linear algebra backends typically treat $L$ as a full $S \times S$ matrix.

**Methodological and Conceptual Complexity** While structurally efficient, this flexibility introduces additional *conceptual and implementation complexity* for the researcher. Unlike standard diagonal approximations which are trivial to integrate, our approach requires the careful management of the rank $R$ and specific stabilization strategies to ensure numerical consistency in the second-order distribution. However, this increased modeling effort is a necessary compromise to achieve joint uncertainty quantification at scales where traditional full-rank methods are physically impossible.

# D Additional Results

## D.1 Qualitative Results

We provide additional qualitative results for every performed tasks. Figures 6 presents optical flow on Flying Chairs, 7 depicts CelebA inpainting, 8 shows CelebA colorization, and 9 illustrates NYU inpainting. Figure 10 compares both, eigenvectors in both random orientations as well as the different used Bayesian methods.

The optical flow visualization of Figure 6 encodes the 2D motion vectors into a color image using a color wheel scheme. Each pixel's hue corresponds to the direction of motion, covering all angles in a circular manner (e.g., red for rightward motion, green for upward, blue for leftward, etc.). The color saturation or intensity represents the magnitude of the motion, with brighter and more saturated colors indicating higher motion speeds. This method allows intuitive interpretation of both the direction and speed of movement in the scene or the direction of the movement uncertainty for the Eigenvectors.

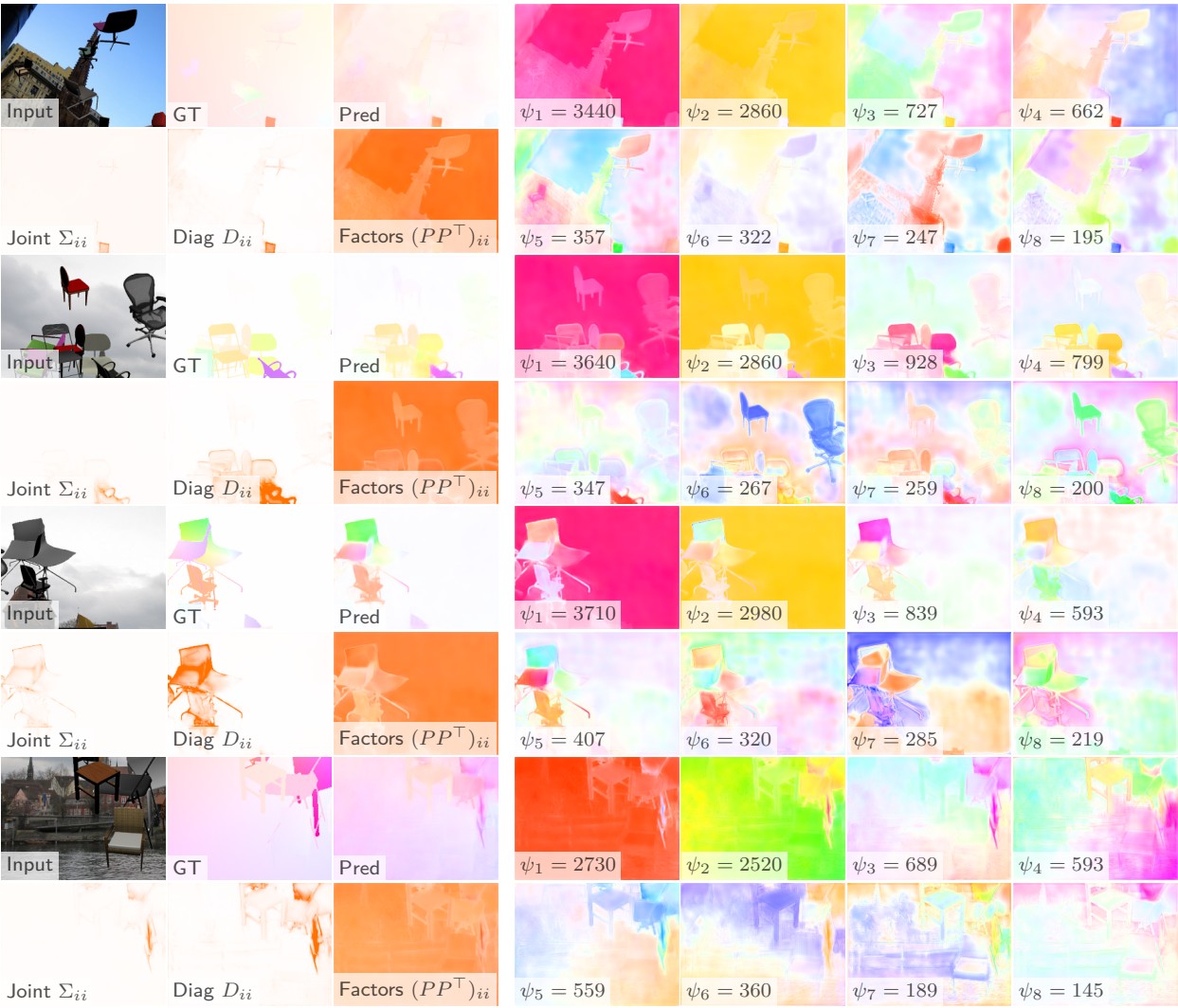

Figure 6: Additional Qualitative Results on Flying Chairs Optical Flow. Random test set samples depicting the input, ground truth (ground truth), prediction (Pred), and parameters of the predictive distribution. For space efficiency, only the first of the two concatenated input frames is visualized. As optical flow is a 2-channel task, the joint low-rank matrix $P$ is visualized using a colorwheel: hue represents the direction of flow uncertainty, while intensity corresponds to the magnitude of the offset. The first three columns show the input, GT, and Pred (top row), and the corresponding marginal variances $(\Sigma_{ii}, D_{ii}, (PP^\top)_{ii})$ (bottom row). Columns 4–7 visualize the 8 most significant eigenvectors of $P$ in descending order of their singular values $\Psi$. We observe that the background often dominates the first two eigenvectors, while the foreground and object boundaries appear in later, more fine-grained factors. Note that the orientation of these singular vectors is arbitrarily chosen; inverting a vector results in complementary colors on the colorwheel, representing the same underlying correlation. These visualizations, uniquely enabled by our covariance modeling, reveal the underlying factors of maximum variability and provide an upper bound on the angles between the eigenvectors of $PP^\top$ and the full covariance $\Sigma$.

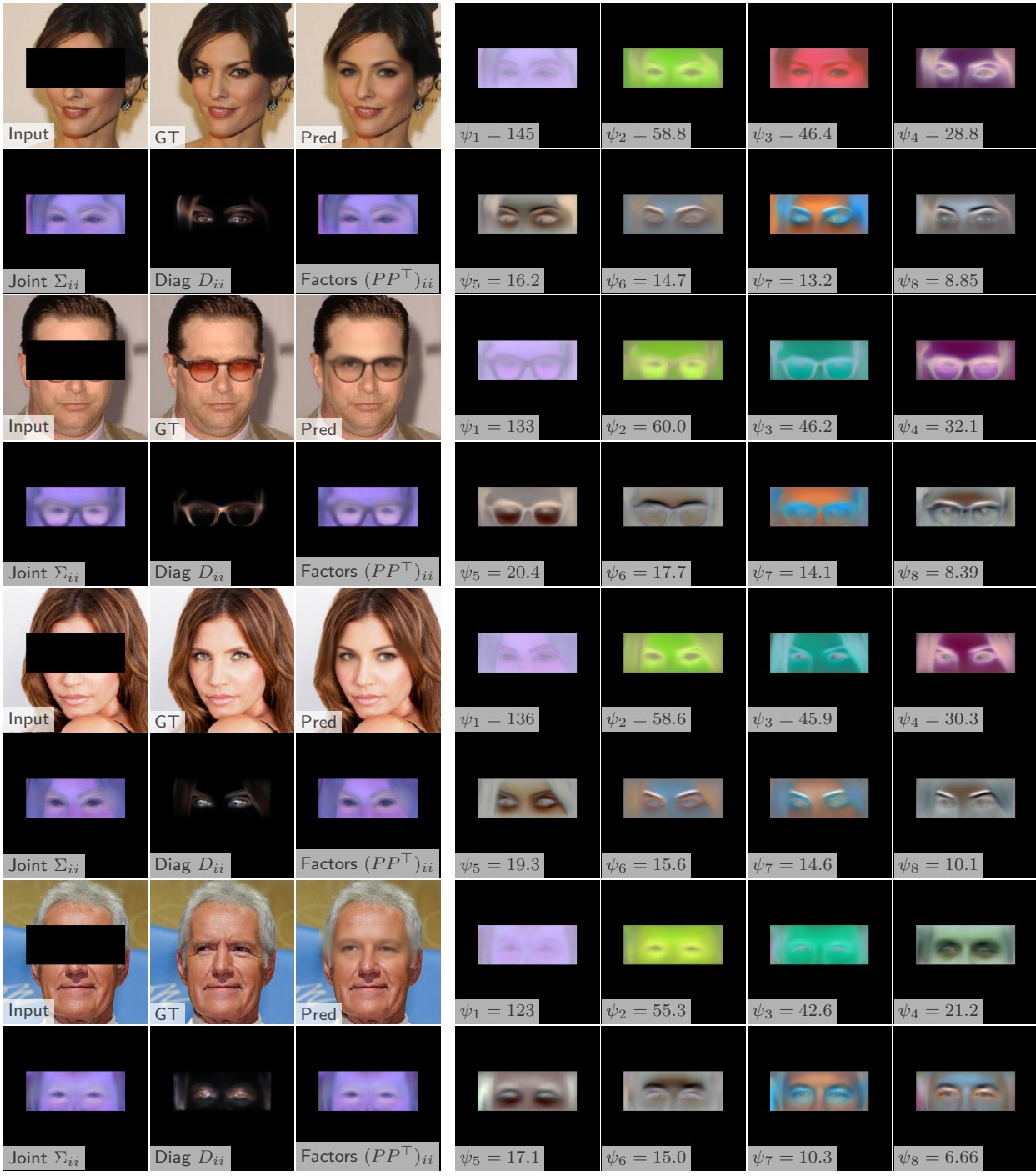

Figure 7: Additional Qualitative Results: CelebA Eye Inpainting. Random test set samples showing the masked input, ground truth (GT), prediction (Pred), and predictive distribution parameters. The bottom row of the first three columns visualizes the marginal variances $\Sigma_{ii}$, $D_{ii}$, and $(PP^\top)_{ii}$. The subsequent columns display the 8 most significant eigenvectors of $P$ in descending order of their singular values $\Psi$. We observe that the primary eigenvectors capture global correlations—such as symmetrical adjustments to both eyes or skin tone—while subsequent factors represent fine-grained, localized details of the iris and eyelids. The singular values quantify the relative importance of these directional correlations, with the first vectors often being an order of magnitude (10×) more significant than the 8th.

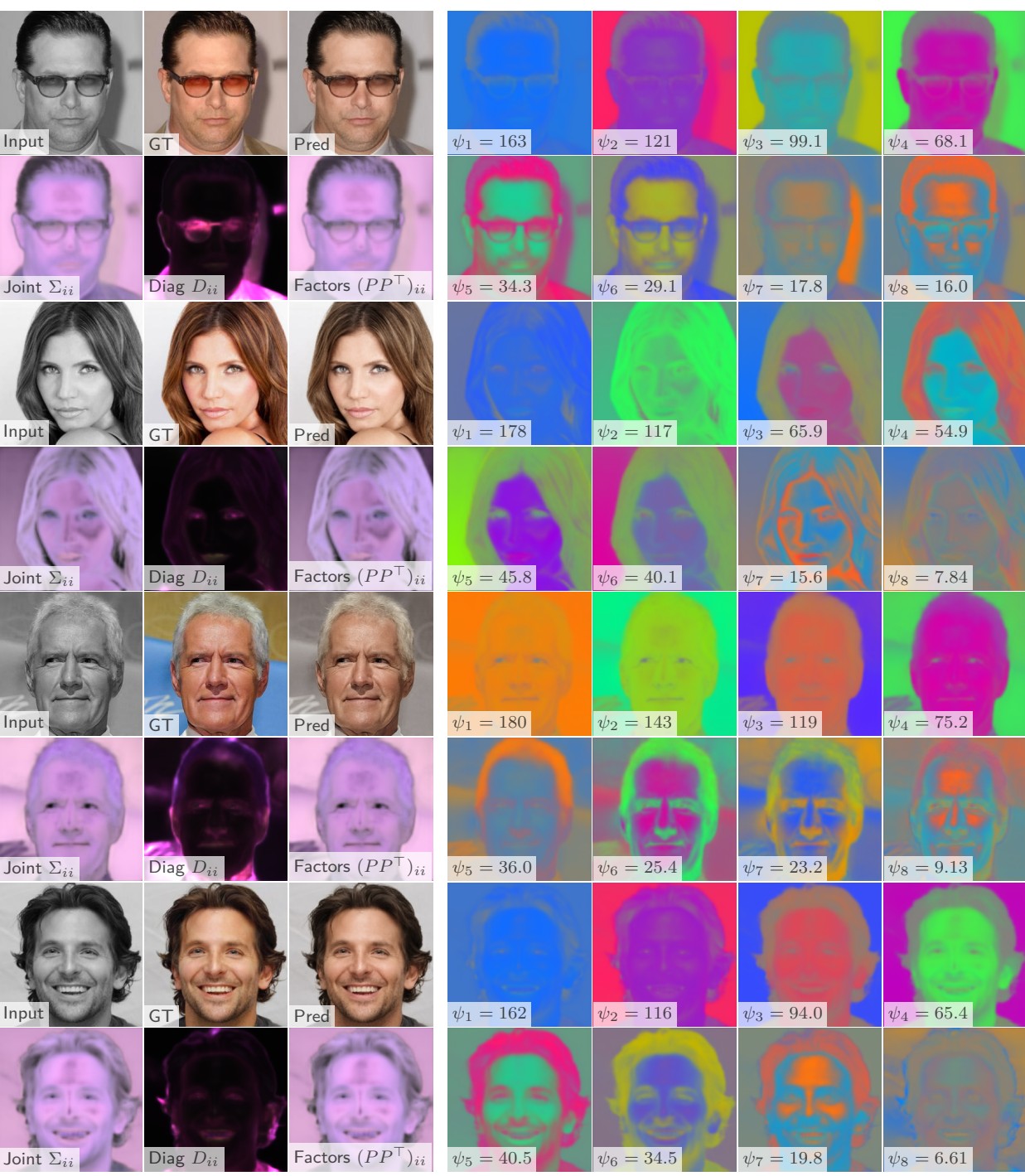

Figure 8: Additional Qualitative Results: CelebA Colorization. Random test set samples showing grayscale input, ground truth (GT), prediction (Pred), and parameters of the predictive distribution. The first three columns visualize the input, ground truth, Pred (top), and the marginal variances $(\Sigma_{ii}, D_{ii}, (PP^\top)_{ii})$ (bottom). The subsequent columns display the 8 most significant eigenvectors of the joint low-rank matrix $P$ in descending order of their singular values $\Psi$. These visualizations reveal a hierarchical structure: the first eigenvectors capture global color temperature shifts (e.g., orange–blue or purple–green axes), while later factors represent increasingly fine-grained and localized features, such as hair color or eye-specific hues. The relative magnitudes of $\Psi$ illustrate the importance of these directional correlations, with the primary factors typically being an order of magnitude ($10\times$) more significant than the 8th.

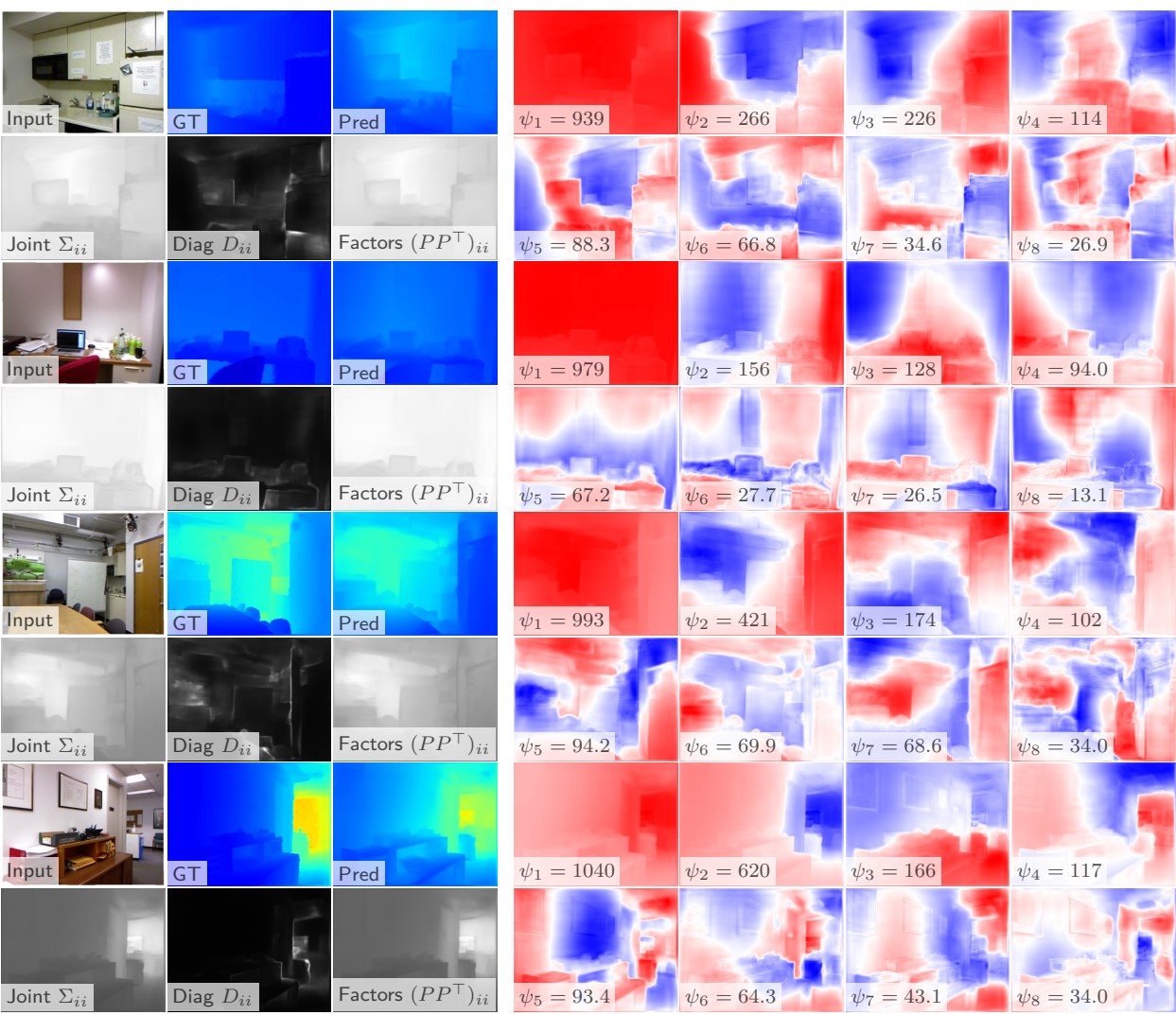

Figure 9: Additional Qualitative Results: NYU Depth Estimation. Random test set samples showing RGB input, ground truth (GT), prediction (Pred), and parameters of the predictive distribution. For single-channel depth, the eigenvectors of the joint low-rank matrix $P$ are visualized using a red–blue diverging colormap to intuitively show positive and negative offsets from the mean. The first three columns visualize the input, GT, Pred (top), and the marginal variances $(\Sigma_{ii}, D_{ii}, (PP^\top)_{ii})$ (bottom). The subsequent columns display the 8 most significant eigenvectors of $P$ in descending order of their singular values $\Psi$. These factors reveal a hierarchical structure: dominant eigenvectors capture global depth shifts or planar tilt, while later factors focus on fine-grained, localized uncertainty around object boundaries and complex geometries. The relative magnitudes of $\Psi$ illustrate the importance of these directional correlations, with the primary factors typically being an order of magnitude ($10\times$) more significant than the 8th.

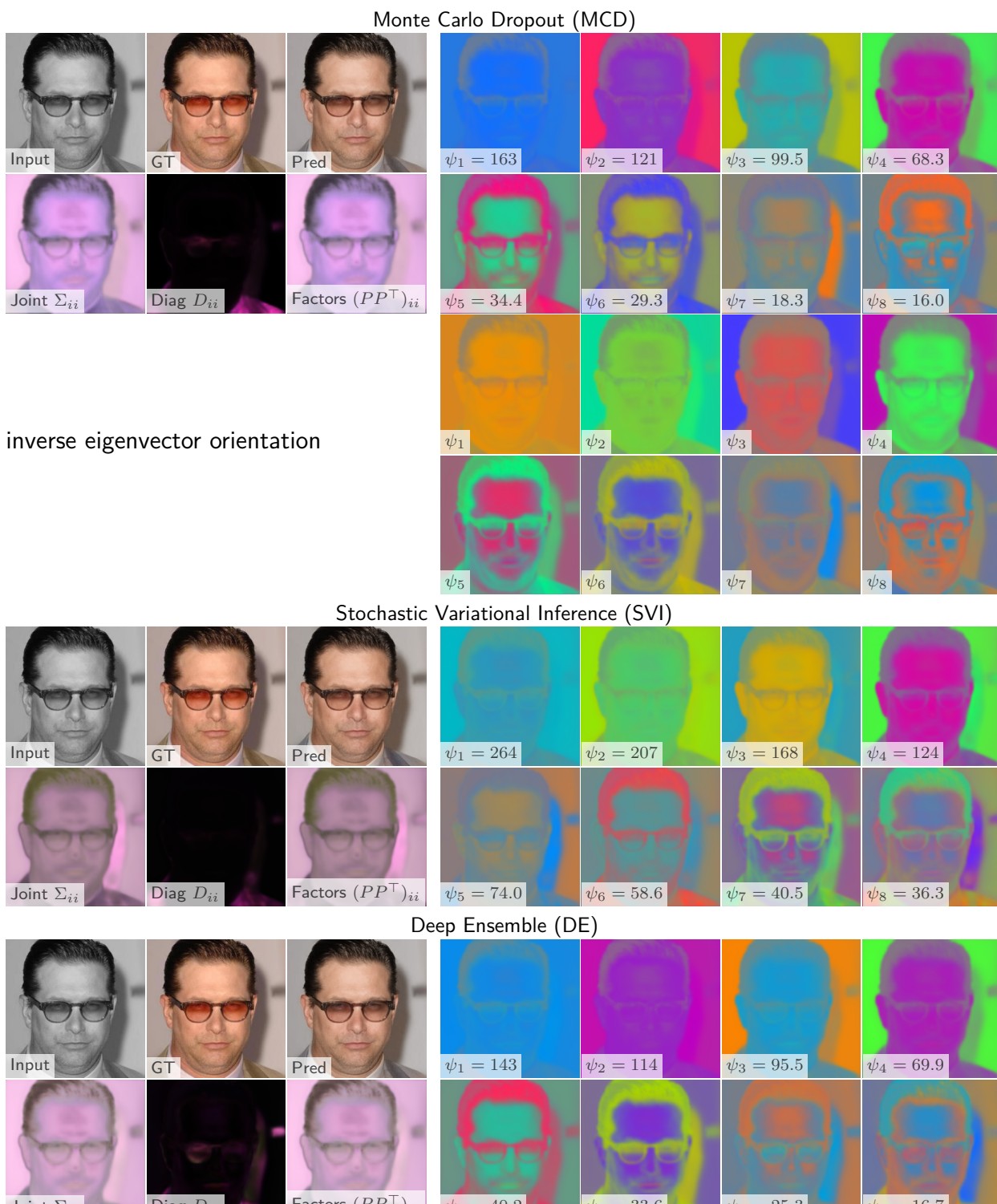

Figure 10: Additional Qualitative Results: Comparison of Bayesian Methods. A random test set sample showing input, ground truth (GT), and predictions across various Bayesian frameworks for CelebA colorization. We visualize the marginal variances and the 8 most significant eigenvectors of $P$ for each method to highlight differences in captured correlation structures. For the primary method, we additionally visualize the eigenvectors with an inverted sign to demonstrate their mathematical equivalence; the choice of orientation is stochastic and does not affect the underlying covariance $PP^\top$.

### D.2  Selective Prediction via Differential Entropy

In this section, we evaluate the utility of our uncertainty estimates for *selective prediction*. A well-calibrated uncertainty metric should allow a system to identify and "abstain" from predictions likely to have high error, thereby improving the average performance on the remaining retained samples.

**Uncertainty Metric: Differential Entropy**    To condense the high-dimensional uncertainty of a prediction into a single scalar for ranking, we employ the **differential entropy** $h$ of the multivariate Gaussian predictive distribution $\mathcal{N}(\mu, \Sigma)$. For an output space of dimension $S$, the entropy is defined as:

$$h(\mathcal{N}(\mu, \Sigma)) = \frac{1}{2} \log \left( (2\pi e)^S |\Sigma| \right) \tag{13}$$

In our LR+D framework, the covariance matrix $\Sigma$ is structured as $\mathbf{D} + \mathbf{V}\mathbf{V}^\top$. We calculate the determinant $|\Sigma|$ efficiently using the **Matrix Determinant Lemma**: $|\mathbf{D} + \mathbf{V}\mathbf{V}^\top| = |\mathbf{I} + \mathbf{V}^\top \mathbf{D}^{-1} \mathbf{V}| \cdot |\mathbf{D}|$. This allows the entropy calculation to account for the structural dependencies captured by the low-rank components, which are ignored by factorized models.

**Evaluation Framework: Coverage–Risk Curves**    We assess the quality of the uncertainty ranking using **Coverage–Risk Curves**, where the risk is defined as the Negative Log-Likelihood (negative log-likelihood (NLL)). These curves visualize the trade-off between the fraction of data the model predicts on and the resulting predictive quality:

- **Coverage ($c$):** The fraction of the test set retained (x-axis). We sort all test samples by their predictive entropy in ascending order. A coverage of $c = 0.1$ includes only the 10% most "confident" predictions according to the model's internal entropy metric.
- **Risk ($R(c)$):** The average NLL calculated on the subset of samples defined by coverage $c$ (y-axis). Unlike error-based metrics, this directly evaluates the probabilistic fit of the model's predicted distribution.

As shown in Figure 11, both the diagonal (D) and LR+D models exhibit a clear downward trend in risk as coverage decreases. This indicates that for both parameterizations, the differential entropy serves as an effective predictor of the NLL. Notably, the LR+D models consistently maintain a lower risk profile across the entire coverage range, demonstrating that the structural correlations captured by our method result in a superior probabilistic fit that persists even as we subset the most certain predictions.

### D.3  Quantitative Prediction Errors

In this section, we report the predictive performance of all considered models to ensure that the improvements in TLL (presented in the main text) are driven by superior uncertainty quantification rather than significantly different reconstruction accuracy. Tables 4 and 5 list the $L_2$ and $L_1$ errors, respectively, for both Bayesian and non-Bayesian configurations.

Results for MCD, SVI, and non-Bayesian (✗) models are reported as the mean and standard deviation across 5 independent model seeds. For DE, we report the performance of a single ensemble (composed of $M = 5$ members) due

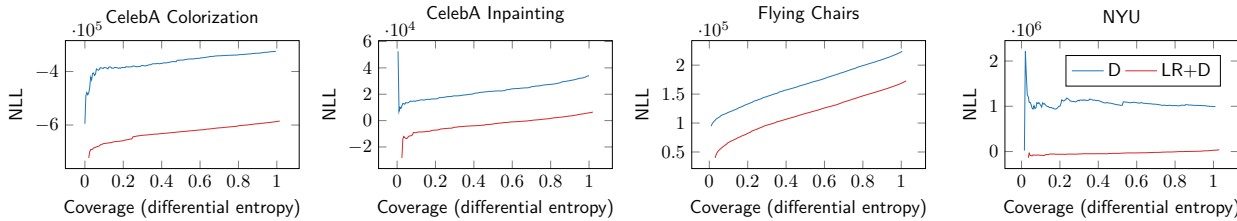

Figure 11: Selective prediction results using differential entropy as the uncertainty score NLL. Coverage (x-axis) shows the fraction of predictions retained when selecting the $1 - \alpha$ lowest-uncertainty predictions, while risk (y-axis) measures the corresponding NLL. Results are shown for diagonal D and LR+D models with MCD across all tasks. For both D and LR+D its according entropy is a good predictor for the NLL .

to the substantial computational cost of training multiple independent ensembles. All Bayesian models are evaluated using $T = 64$ weight samples.

| Param. | $R_W$ | Epistemic | CelebA | | Flying Chairs | NYU |
| | | | Inpainting | Colorization | Optical Flow | Depth |
|---|---|---|---|---|---|---|
| M | 0 | ✗ | 0.0470 | 0.0965 | 5.50 | 0.810 |
| D | 0 | ✗ | $0.0478 \pm 0.0006$ | $0.0979 \pm 0.0008$ | $5.57 \pm 0.06$ | $0.826 \pm 0.015$ |
| D | 0 | MCD | $0.0476 \pm 0.0008$ | $0.0934 \pm 0.0010$ | $5.56 \pm 0.09$ | $0.804 \pm 0.019$ |
| | | SVI | $0.0470 \pm 0.0005$ | $0.0969 \pm 0.0005$ | $5.62 \pm 0.07$ | $0.831 \pm 0.020$ |
| | | DE | 0.0457 | 0.0900 | **5.26** | 0.753 |
| LR+D | 4 | MCD | $0.0454 \pm 0.0001$ | $0.0923 \pm 0.0005$ | $5.58 \pm 0.05$ | $0.750 \pm 0.009$ |
| | 8 | ✗ | $0.0455 \pm 0.0002$ | $0.0961 \pm 0.0007$ | $5.49 \pm 0.03$ | $0.791 \pm 0.013$ |
| | | MCD | $0.0451 \pm 0.0001$ | $0.0924 \pm 0.0007$ | $5.56 \pm 0.05$ | $0.772 \pm 0.023$ |
| | | SVI | $0.0449 \pm 0.0001$ | $0.0969 \pm 0.0019$ | $5.58 \pm 0.11$ | $0.810 \pm 0.024$ |
| | | DE | **0.0448** | **0.0887** | 5.28 | **0.746** |
| | 12 | MCD | $0.0452 \pm 0.0001$ | $0.0917 \pm 0.0006$ | $5.61 \pm 0.06$ | $0.749 \pm 0.009$ |
| | 16 | MCD | $0.0453 \pm 0.0001$ | $0.0915 \pm 0.0001$ | $5.54 \pm 0.07$ | $0.774 \pm 0.020$ |

Table 4: $L_2$ prediction errors across different dataset-task combinations. Values for ✗, MCD, and SVI represent mean $\pm$ standard deviation over 5 seeds. DE results are from a single ensemble.

| Param. | $R_W$ | Epistemic | CelebA | | Flying Chairs | NYU |
| | | | Inpainting | Colorization | Optical Flow | Depth |
|---|---|---|---|---|---|---|
| M | 0 | ✗ | 0.0327 | 0.0639 | 2.50 | 0.630 |
| D | 0 | ✗ | $0.0333 \pm 0.0004$ | $0.0644 \pm 0.0003$ | $2.60 \pm 0.09$ | $0.645 \pm 0.016$ |
| D | 0 | MCD | $0.0332 \pm 0.0006$ | $0.0606 \pm 0.0008$ | $2.58 \pm 0.08$ | $0.627 \pm 0.012$ |
| | | SVI | $0.0327 \pm 0.0004$ | $0.0631 \pm 0.0004$ | $2.63 \pm 0.07$ | $0.653 \pm 0.021$ |
| | | DE | 0.0317 | 0.0581 | **2.33** | 0.589 |
| LR+D | 4 | MCD | $0.0315 \pm 0.0001$ | $0.0602 \pm 0.0004$ | $2.59 \pm 0.06$ | $0.583 \pm 0.006$ |
| | 8 | ✗ | $0.0315 \pm 0.0002$ | $0.0630 \pm 0.0004$ | $2.50 \pm 0.04$ | $0.620 \pm 0.011$ |
| | | MCD | $0.0313 \pm 0.0001$ | $0.0600 \pm 0.0006$ | $2.57 \pm 0.04$ | $0.600 \pm 0.018$ |
| | | SVI | $0.0312 \pm 0.0002$ | $0.0644 \pm 0.0014$ | $2.58 \pm 0.11$ | $0.639 \pm 0.024$ |
| | | DE | **0.0311** | **0.0574** | 2.35 | **0.580** |
| | 12 | MCD | $0.0314 \pm 0.0001$ | $0.0596 \pm 0.0003$ | $2.60 \pm 0.07$ | $0.582 \pm 0.008$ |
| | 16 | MCD | $0.0316 \pm 0.0001$ | $0.0598 \pm 0.0004$ | $2.54 \pm 0.08$ | $0.609 \pm 0.020$ |

Table 5: $L_1$ prediction errors across different dataset–task combinations. Values for ✗, MCD, and SVI represent mean $\pm$ standard deviation over 5 seeds. DE results are from a single ensemble.

### D.4 Additional Ablation Study

**Test set variabiltiy** While the main results in Table 1 report the standard deviation across the samples within the testset to demonstrate training stability, this provides little insight into how the model performs on an individual image level. In this section, we analyze the per-sample variability across the entire test set.

As shown in Table 6, the standard deviation here represents the fluctuation in TLL caused by varying image complexity and task ambiguity. High variability in tasks like Modified National Institute of Standards and Technology database (MNIST) or CelebA colorization suggests that while the model performs well on average, certain "outlier" samples (e.g., highly occluded regions or unusual lighting) result in significantly lower likelihoods. This variance highlights the heteroscedastic nature of the tasks, which our LR+D approach is specifically designed to capture through the joint covariance matrix P. The relatively high variance in test log-likelihood on MNIST can be attributed to the discrete and multimodal nature of the task: predictions that capture the correct digit mode yield substantially higher likelihood than predictions corresponding to an incorrect digit, leading to large per-sample differences in log-likelihood.

**Comparison with Nehme et al. (2024)** We trained additional models to compare with the approach of Nehme et al. (2024), as summarized in Tables 7, 8, and 4. To enable the use of Nehme's Neural Posterior Principal Components (NPPC) loss for the covariance factor, several modifications were necessary.

| Paramet. | Epist. | Method | CelebA | | Flying Chairs | NYU |
| | | | Inpainting $\times 100$ | Colorization $\times 1000$ | Optical Flow $\times 1000$ | Depth $\times 1000$ |
|---|---|---|---|---|---|---|
| D | ✗ | | -471 $\pm$ 617 | 240 $\pm$ 554 | -231 $\pm$ 118 | -2944 $\pm$ 4817 |
| LR+D | ✗ | | -513 $\pm$ 1031 | 495 $\pm$ 433 | -184 $\pm$ 133 | -1613 $\pm$ 2868 |
| D | MCD | | -341 $\pm$ 551 | 324 $\pm$ 321 | -224 $\pm$ 83 | -997 $\pm$ 1970 |
| | SVI | | -439 $\pm$ 493 | 340 $\pm$ 259 | -227 $\pm$ 115 | -655 $\pm$ 780 |
| | DE | | -249 $\pm$ 402 | 374 $\pm$ 159 | -213 $\pm$ 72 | -332 $\pm$ 492 |
| LR+D | MCD | $\mathbb{E}[W]$ | -120 $\pm$ 478 | 565 $\pm$ 203 | -170 $\pm$ 86 | -176 $\pm$ 584 |
| | SVI | $\mathbb{E}[W]$ | -245 $\pm$ 481 | 558 $\pm$ 181 | **-162** $\pm$ 104 | -346 $\pm$ 690 |
| | MCD | TSVD | -65 $\pm$ 245 | 585 $\pm$ 94 | -173 $\pm$ 72 | -36 $\pm$ 274 |
| | SVI | TSVD | -218 $\pm$ 335 | 585 $\pm$ 108 | -166 $\pm$ 93 | -195 $\pm$ 447 |
| | DE | TSVD | **-50** $\pm$ 220 | **589** $\pm$ 97 | -172 $\pm$ 67 | **-13** $\pm$ 210 |

Table 6: Test Set Variability. We evaluate the mean TLL and the standard deviation calculated across all individual samples in the test set for a single representative model seed. In contrast to Table 1, these values reflect the diversity and difficulty of the dataset rather than model initialization. The high variance observed in MNIST and depth estimation tasks quantifies the existence of "hard" samples where the predictive distribution exhibits significantly higher uncertainty or multi-modality.

First, to improve training stability, the mean and uncertainty predictions were separated into two models, with the uncertainty model optionally receiving the mean model's output as an additional input (indicated by the *Combined* column: ✓ vs. ✗). Second, Gram–Schmidt orthogonalization was applied to enforce orthogonality of the covariance directions (*Gram–Schmidt* column: ✓ vs. ✗), a prerequisite for the NPPC loss. Finally, the NPPC loss was applied to the covariance factor $P$, while the diagonal covariance terms remained optimized with the LR+D loss using a stop-gradient on $P$.

Due to instability, NPPC training was only viable with separate models and Gram–Schmidt projection; combined mean–uncertainty models with NPPC were not stable. Thus, we present results showing a stepwise transition from the baseline LR+D model to the NPPC configuration.

Before any subsequent loss computation or evaluation, the raw low-rank model output matrix $P^W$ consists of columns $p_r^W$ that represent initial (non-orthogonal) low-rank directions. These columns are first orthogonalized using the Gram–Schmidt process to obtain the orthogonalized directions $p_r^a$. Normalizing these columns produces the orthonormal directions $\bar{p}_r$.

In matrix form, these are arranged as:

$$P^W = \begin{bmatrix} p_1^W & p_2^W & \cdots & p_R^W \end{bmatrix}, \quad P^\star = \begin{bmatrix} p_1^\star & p_2^\star & \cdots & p_R^\star \end{bmatrix}, \quad \bar{P} = \begin{bmatrix} \bar{p}_1 & \bar{p}_2 & \cdots & \bar{p}_R \end{bmatrix}.$$

Each $p_r^W, p_r^\star, \bar{p}_r$ is a column vector in these respective matrices representing different stages of processing for the basis directions in the low-rank decomposition. The NPPC objectives are calculated using the detached prediction error $e(x_i) = \lfloor \mu(x_i) - y_i \rfloor$, and decompose into direction and variance losses:

$$\mathcal{L}_{\bar{P}} = 1 - \sum_i \left\| \bar{P}^\top(x_i) e(x_i) \right\|^2$$

$$\mathcal{L}_{P^\star} = \sum_i \underbrace{\frac{1}{\|e(x_i)\|^4}}_{\text{normalization}} \sum_{r=1}^{R^W} \left( \|p_r^\star(x_i)\|^2 - \left( \bar{p}_r^\top(x_i) e(x_i) \right)^2 \right)^2$$

Whereas the normalization $\frac{1}{\|e(x_i)\|^4}$, is not explicitly mentioned in Nehme et al. (2024) while the accompanying implementation of the NPPC actually normalizes the variance loss $L_{P^\star}$. Therefore, we stick with their implementation and adapt the formulas accordingly.

---

**Algorithm 1** Gram-Schmidt

---

**Require:** Model output vectors $p_1^W, p_2^W, \ldots, p_R^W$
**Ensure:** Orthogonal vectors $p_1^\star, p_2^\star, \ldots, p_R^\star$
**Ensure:** Orthonormal vectors $\bar{p}_1, \bar{p}_2, \ldots, \bar{p}_R$

1: $p_1^\star = p_1^W$

2: $\bar{p}_1 = \dfrac{p_1^\star}{\|p_1^\star\|}$

3: **for** $k = 2$ **to** $K$ **do**

4:     Form the matrix of previous orthonormal columns:

$$\bar{P}^{(r-1)} = \begin{bmatrix} \bar{p}_1 & \cdots & \bar{p}_{r-1} \end{bmatrix}$$

5:     Orthogonalize:

$$p_r^\star = p_r^W - \bar{P}^{(r-1)} \left( (\bar{P}^{(r-1)})^\top p_r^\star \right)$$

6:     Normalize:

$$\bar{p}_r = \frac{p_r^\star}{\|p_r^\star\|}$$

7: **end for**

---

| Combined | Gram Schmid | Loss | CelebA Inpainting $\times 100$ | CelebA Colorization $\times 1000$ | Flying Chairs Optical Flow $\times 1000$ |
|---|---|---|---|---|---|
| ✓ | ✗ | | **-65** $\pm$ 239 | **585** $\pm$ 116 | -173 $\pm$ 73 |
| | ✓ | | -91 $\pm$ 341 | 582 $\pm$ 113 | -167 $\pm$ 75 |
| ✗ | ✗ | | -79 $\pm$ 162 | 566 $\pm$ 58 | **-91** $\pm$ 132 |
| | ✓ | | -75 $\pm$ 156 | 565 $\pm$ 64 | -95 $\pm$ 120 |
| | | NPPC | -96 $\pm$ 284 | 571 $\pm$ 99 | -100 $\pm$ 206 |

Table 7: TLL for each model configuration across tasks. The table compares the impact of combined versus separate mean and uncertainty models (*Combined*), application of Gram–Schmidt orthogonalization (*Gram Schmid*), and different loss functions, including NPPC. Values are reported as mean $\pm$ standard deviation, scaled as indicated.

The joint loss becomes:

$$\mathcal{L}_{\mathrm{lrd}\star} = \sum_i \log \mathcal{N} \left( y_i \mid \lfloor \mu(x_i) \rfloor, D(x_i) + \lfloor P(x_i)P^\top(x_i) \rfloor \right) \tag{14}$$

$$\mathcal{L} = \alpha \, \mathcal{L}_{\mathrm{lrd}\star} + \beta \, \mathcal{L}_{\bar{P}} + \beta \, \mathcal{L}_{p^\star} \tag{15}$$

We evaluate models using TLL (Table 7) and relative reconstruction error

$$e = \mu(x) - y, \quad \frac{e - PP^\top e}{e},$$

which quantifies the fraction of residual error unexplained by the covariance directions (Table 8). Prediction errors from the previous section (Table 4) serve as baseline references.

The LR+D-trained models consistently outperform NPPC variants in TLL, indicating better likelihood-based prediction. Conversely, NPPC reduces the relative reconstruction error, suggesting improved capture of residual structure by the covariance factor. These results highlight a trade-off: optimizing for TLL favors LR+D models, while NPPC better models orthogonal residual errors, guiding metric-dependent optimization choices.

Models without combined prediction use the same mean model as the first row of Table 4, allowing direct reading of their prediction errors from that reference.

| Combined | Gram Schmid | Loss | CelebA | | Flying Chairs |
| | | | Inpainting | Colorization | Optical Flow |
|---|---|---|---|---|---|
| ✓ | ✗ | | $\mathbf{0.36} \pm 0.13$ | $0.63 \pm 0.08$ | $0.62 \pm 0.13$ |
| | ✓ | | $\mathbf{0.36} \pm 0.13$ | $0.62 \pm 0.08$ | $0.63 \pm 0.13$ |
| ✗ | ✗ | | $0.37 \pm 0.13$ | $0.68 \pm 0.08$ | $0.62 \pm 0.14$ |
| | ✓ | | $0.37 \pm 0.13$ | $0.67 \pm 0.08$ | $0.61 \pm 0.14$ |
| | | NPPC | $\mathbf{0.36} \pm 0.12$ | $\mathbf{0.61} \pm 0.07$ | $\mathbf{0.53} \pm 0.13$ |

Table 8: Relative reconstruction error $\left( \frac{e - PP^\top e}{e} \right)$ for each model configuration on the same test sets. This metric captures the proportion of the prediction residual error not explained by the covariance model directions, providing insight into the quality of uncertainty estimation. Column organization matches Table 7 for direct comparison.

**Joint vs. separate training of mean and uncertainty models**  While the design choice to jointly train mean prediction and uncertainty estimation models is conceptually cleaner and yields superior results on three out of four benchmark tasks, an exception arises in the Flying Chairs dataset. For this dataset, the separate model setup performs significantly better. In the split configuration, the uncertainty model receives as input the concatenation of both the original input and the output of the mean prediction model.

This architectural difference may explain the observed discrepancy in performance. Unlike image inpainting and colorization tasks—where the model has no direct way to verify the correctness of its predictions—the Flying Chairs task inherently allows the uncertainty model to implicitly "check" prediction accuracy. Specifically, the uncertainty model can leverage the input images and predicted flow outputs to learn pixel shifts that align with ground truth, effectively evaluating the prediction quality.

We hypothesize that this direct feedback mechanism enables the separate uncertainty model in Flying Chairs to better estimate prediction errors, whereas joint training suffices or outperforms for other tasks where such verification is unavailable or less direct. This is supported by the negative log-likelihood comparisons reported in Table 7, where the split model outperforms the combined model specifically on the Flying Chairs dataset.

This suggests that task-specific characteristics and the degree of prediction observability should guide the choice between joint and separated uncertainty modeling architectures.

**Retaining variance of TSVD**  One component of our proposed method is to retain the variance of the removed columns after dimensionality reduction using SVD, see 9. In Table 9 we ablate this design choice. The column $\hat{D}$ indicates whether the diagonal $D$ is updated (✓) after performing SVD according to Equation 9, or if the original $D$ is retained (✗) as per Equation 6. Our ablation shows that updating the diagonal $D$ appears to slightly improve the average, TLL while also enhancing prediction consistency and reducing test set variability. This is consistent for three different configurations of dimensionality reductions, see Table 9.

**Influence of output layer columns**  Furthermore, Table 10 evaluates the impact of the number of columns predicted by the model's output layer. Increasing the number of columns benefits the TLL for the 3 tasks with larger images and therefore a higher dimensional output space (CelebA inpainting and colorization as well as optical flow estimation on flying chairs). However, this increases computational complexity and lead to numerical instabilities during training, as models with $R^W \geq 20$ columns fail for all tasks. Balancing these trade-offs, we chose a rank of 8, which aligns close with the choice of Monteiro et al. (2020).

Finally, Table 11 presents an extended ablation of various parameters, non-Bayesian with various Bayesian Models (epis) and both kinds of distribution parametrizations (Param). Therefore, it compares a purely diagonal (D) uncertainty with our LR+D parametrization. The representation took $T$ Bayesian samples, and results in $R$ columns of the low-rank matrix. The aleatoric matrix $P^a$ is in some rows estimated using the expected weights $\mathbb{E}[W]$ approximation. The number of columns of the matrix ($P$) is optionally reduced using TSVD. The matrix is omitted for purely diagonal D Parameterizations (-), and is either truncated (✓) or kept (✗) for LR+D models. The column $\hat{D}$ indicates whether the diagonal $D$ is updated (✓) after performing TSVD according to Equation 9, or if the original $D$ is retained as per Equation 6, despite the dimensionality reduction of $P$.

| $R$ | $\hat{D}$ | CelebA | | Flying Chairs | NYU |
| | | Inpainting $\times 100$ | Colorization $\times 1000$ | Optical Flow $\times 1000$ | Depth $\times 1000$ |
|---|---|---|---|---|---|
| 8 | ✓ | -194 ± 281 | 521 ± 138 | -194 ± 84 | -967 ± 1773 |
| | ✗ | -294 ± 556 | 519 ± 241 | -186 ± 109 | -182 ± 440 |
| 16 | ✓ | -152 ± 263 | 548 ± 139 | -186 ± 82 | -451 ± 1018 |
| | ✗ | -211 ± 439 | 550 ± 203 | -178 ± 98 | -143 ± 405 |
| 32 | ✓ | -109 ± 249 | 567 ± 129 | -179 ± 76 | -218 ± 584 |
| | ✗ | -143 ± 368 | 571 ± 171 | -172 ± 87 | -85 ± 313 |
| 64 | ✓ | -65 ± 239 | 585 ± 116 | -173 ± 73 | -87 ± 352 |
| | ✗ | -82 ± 311 | 589 ± 141 | -167 ± 80 | -51 ± 274 |
| 128 | ✓ | -22 ± 223 | 602 ± 87 | -167 ± 72 | -8 ± 219 |
| | ✗ | -28 ± 253 | 604 ± 96 | -163 ± 75 | 7 ± 187 |
| 256 | ✓ | 17 ± 201 | 616 ± 75 | -161 ± 71 | 37 ± 151 |
| | ✗ | 16 ± 207 | 617 ± 78 | -160 ± 72 | 40 ± 144 |
| 512 | ✓ | 39 ± 179 | 626 ± 68 | -158 ± 70 | 63 ± 113 |
| | ✗ | 39 ± 179 | 626 ± 68 | -158 ± 70 | **63** ± 113 |
| 576 | - | **40** ± 177 | **627** ± 67 | **-158** ± 70 | 61 ± 123 |

Table 9: Comparison between adapting the diagonal $D$ after performing the SVD according to Equation 9 or not. Here $R$ denotes the number of columns in the resulting representation. The columns $P^a$ and $P^e$ denote if and to what degree the dimensionality is reduced after sampling using SVD. The numbers in brackets denote the kept singular vectors, which result in columns $R$. The column $\hat{D}$ indicates whether the diagonal $D$ is updated (✓) after performing SVD according to Equation 9, or if the original $D$ is retained as per Equation 6, despite the dimensionality reduction of $P$. We show in the first three rows that updating the diagonal $D$ appears to slightly improve the average TLL while also enhancing prediction consistency and reducing test set variability.

| | | CelebA | | Flying Chairs | NYU |
|---|---|---|---|---|---|
| $R$ | $R_W$ | Inpainting $\times 100$ | Colorization $\times 1000$ | Optical Flow $\times 1000$ | Depth $\times 1000$ |
| 16 | 4 | -218 $\pm$ 429 | 537 $\pm$ 140 | **-180** $\pm$ 90 | -180 $\pm$ 419 |
| | 8 | 96 $\pm$ 88 | **548** $\pm$ 139 | -186 $\pm$ 82 | -143 $\pm$ 405 |
| | 12 | 103 $\pm$ 80 | 526 $\pm$ 76 | -190 $\pm$ 70 | **-11** $\pm$ 76 |
| | 16 | **-129** $\pm$ 155 | 535 $\pm$ 72 | -192 $\pm$ 71 | -15 $\pm$ 89 |
| 32 | 4 | -168 $\pm$ 398 | 556 $\pm$ 118 | **-174** $\pm$ 85 | -117 $\pm$ 353 |
| | 8 | -109 $\pm$ 249 | 567 $\pm$ 129 | -179 $\pm$ 76 | -85 $\pm$ 313 |
| | 12 | **-98** $\pm$ 229 | 555 $\pm$ 87 | -182 $\pm$ 68 | **3** $\pm$ 88 |
| | 16 | -102 $\pm$ 167 | **568** $\pm$ 76 | -182 $\pm$ 70 | -8 $\pm$ 117 |
| 64 | 4 | -114 $\pm$ 348 | 573 $\pm$ 100 | **-167** $\pm$ 82 | -53 $\pm$ 268 |
| | 8 | -65 $\pm$ 239 | 585 $\pm$ 116 | -173 $\pm$ 73 | -51 $\pm$ 274 |
| | 12 | **-58** $\pm$ 224 | 582 $\pm$ 81 | -174 $\pm$ 68 | -38 $\pm$ 234 |
| | 16 | -65 $\pm$ 177 | **594** $\pm$ 72 | -172 $\pm$ 71 | **-21** $\pm$ 186 |
| 128 | 4 | -66 $\pm$ 299 | 589 $\pm$ 86 | **-161** $\pm$ 79 | -16 $\pm$ 229 |
| | 8 | -22 $\pm$ 223 | 602 $\pm$ 87 | -167 $\pm$ 72 | **7** $\pm$ 187 |
| | 12 | **-14** $\pm$ 215 | 607 $\pm$ 67 | -166 $\pm$ 69 | -1 $\pm$ 202 |
| | 16 | -21 $\pm$ 184 | **617** $\pm$ 68 | -162 $\pm$ 72 | -42 $\pm$ 297 |

Table 10: Comparison between different number of columns used during training the model. While $R^W$ denotes the number of columns produced by the model without sampling., $R$ denotes the number of columns in the resulting representation. The column $P$ show how SVD is used dimensionality is reduced. The number in the brackets denote the kept singular vectors, which result as columns $R$. The results tend to be better with a higher number of learned columns $R^W$. In general, increasing $R^W$ can cause increasing training time. However, we also experienced instabilities during training for 3 of those tasks when using $R^W = 32$.

Table 11:

| Epis | Param | $R^W$ | $T$ | $P^a$ | TSVD | $\hat{D}$ | $R$ | CelebA Inpainting ×100 | Colorization ×1000 | Flying Chairs Optical Flow ×1000 | NYU Depth ×1000 |
|---|---|---|---|---|---|---|---|---|---|---|---|
| ✗ | D | 0 | 0 + 1 | $\mathbb{E}[W]$ | - | - | 0 | 471 ± 558 | -240 ± 519 | 231 ± 110 | 2944 ± 4355 |
| ✗ | LR+D | 8 | 0 + 1 | $\mathbb{E}[W]$ | ✗ | - | 8 | 513 ± 1104 | -495 ± 267 | 184 ± 119 | 1613 ± 2673 |
| MCD | D | 0 | 64 + 0 | - | - | - | 0 | 341 ± 495 | -324 ± 249 | 224 ± 85 | 997 ± 1494 |
| MCD | D | 0 | 64 + 1 | $\mathbb{E}[W]$ | - | - | 0 | 362 ± 530 | -317 ± 265 | 224 ± 90 | 1018 ± 1543 |
| MCD | LR+D | 4 | 64 + 0 | - | ✓ | ✗ | 4 | 373 ± 652 | -456 ± 272 | 196 ± 125 | 1384 ± 2452 |
| MCD | LR+D | 4 | 64 + 0 | - | ✓ | ✓ | 4 | 298 ± 472 | -478 ± 163 | 194 ± 97 | 330 ± 619 |
| MCD | LR+D | 4 | 64 + 0 | - | ✓ | ✗ | 8 | 317 ± 609 | -506 ± 202 | 186 ± 113 | 667 ± 1223 |
| MCD | LR+D | 4 | 64 + 0 | - | ✓ | ✓ | 8 | 260 ± 452 | -514 ± 152 | 187 ± 94 | 253 ± 523 |
| MCD | LR+D | 4 | 64 + 0 | - | ✓ | ✗ | 16 | 259 ± 558 | -534 ± 168 | 178 ± 104 | 380 ± 776 |
| MCD | LR+D | 4 | 64 + 0 | - | ✓ | ✓ | 16 | 218 ± 429 | -537 ± 140 | 180 ± 90 | 180 ± 419 |
| MCD | LR+D | 4 | 64 + 0 | - | ✓ | ✗ | 32 | 193 ± 481 | -556 ± 134 | 171 ± 95 | 206 ± 524 |
| MCD | LR+D | 4 | 64 + 0 | - | ✓ | ✓ | 32 | 168 ± 398 | -556 ± 118 | 174 ± 85 | 117 ± 353 |
| MCD | LR+D | 4 | 64 + 0 | - | ✓ | ✗ | 64 | 126 ± 390 | -574 ± 108 | 165 ± 87 | 91 ± 348 |
| MCD | LR+D | 4 | 64 + 0 | - | ✓ | ✓ | 64 | 114 ± 348 | -573 ± 100 | 167 ± 82 | 60 ± 285 |
| MCD | LR+D | 4 | 64 + 0 | - | ✓ | ✗ | 128 | 69 ± 312 | -589 ± 89 | 160 ± 81 | 22 ± 242 |
| MCD | LR+D | 4 | 64 + 0 | - | ✓ | ✓ | 128 | 66 ± 299 | -589 ± 86 | 161 ± 79 | 16 ± 229 |
| MCD | LR+D | 4 | 64 + 0 | - | ✓ | ✗ | 256 | 30 ± 257 | -601 ± 77 | 156 ± 76 | -26 ± 167 |
| MCD | LR+D | 4 | 64 + 0 | - | ✓ | ✓ | 256 | 30 ± 256 | -601 ± 77 | 156 ± 76 | -26 ± 167 |
| MCD | LR+D | 8 | 8 + 0 | - | ✗ | - | 72 | 139 ± 355 | -576 ± 150 | 173 ± 90 | 207 ± 497 |
| MCD | LR+D | 8 | 16 + 0 | - | ✗ | - | 144 | 63 ± 268 | -596 ± 114 | 167 ± 80 | 71 ± 305 |
| MCD | LR+D | 8 | 24 + 0 | - | ✗ | - | 216 | 23 ± 232 | -607 ± 92 | 164 ± 75 | 15 ± 227 |
| MCD | LR+D | 8 | 32 + 0 | - | ✗ | - | 288 | -0 ± 211 | -614 ± 83 | 162 ± 73 | -16 ± 181 |
| MCD | LR+D | 8 | 40 + 0 | - | ✗ | - | 360 | -16 ± 197 | -618 ± 77 | 161 ± 72 | -35 ± 152 |
| MCD | LR+D | 8 | 48 + 0 | - | ✗ | - | 432 | -26 ± 188 | -622 ± 73 | 159 ± 71 | -48 ± 133 |
| MCD | LR+D | 8 | 56 + 0 | - | ✗ | - | 504 | -34 ± 182 | -625 ± 69 | 159 ± 71 | -58 ± 121 |
| MCD | LR+D | 8 | 64 + 0 | - | ✓ | ✗ | 8 | 294 ± 556 | -519 ± 241 | 186 ± 109 | 967 ± 1773 |
| MCD | LR+D | 8 | 64 + 0 | - | ✓ | ✓ | 8 | 194 ± 281 | -521 ± 138 | 194 ± 84 | 182 ± 440 |
| MCD | LR+D | 8 | 64 + 0 | - | ✓ | ✗ | 16 | 211 ± 439 | -550 ± 203 | 178 ± 98 | 451 ± 1018 |
| MCD | LR+D | 8 | 64 + 0 | - | ✓ | ✓ | 16 | 152 ± 263 | -548 ± 139 | 186 ± 82 | 143 ± 405 |
| MCD | LR+D | 8 | 64 + 0 | - | ✓ | ✗ | 32 | 143 ± 368 | -571 ± 171 | 172 ± 87 | 218 ± 584 |
| MCD | LR+D | 8 | 64 + 0 | - | ✓ | ✓ | 32 | 109 ± 249 | -567 ± 129 | 179 ± 76 | 85 ± 313 |
| MCD | LR+D | 8 | 64 + 0 | - | ✓ | ✗ | 64 | 82 ± 311 | -589 ± 141 | 167 ± 80 | 87 ± 352 |
| MCD | LR+D | 8 | 64 + 0 | - | ✓ | ✓ | 64 | 65 ± 239 | -585 ± 116 | 173 ± 73 | 36 ± 245 |
| MCD | LR+D | 8 | 64 + 0 | - | ✓ | ✗ | 128 | 28 ± 253 | -604 ± 96 | 163 ± 75 | 8 ± 219 |
| MCD | LR+D | 8 | 64 + 0 | - | ✓ | ✓ | 128 | 22 ± 223 | -602 ± 87 | 167 ± 72 | -7 ± 187 |
| MCD | LR+D | 8 | 64 + 0 | - | ✓ | ✗ | 256 | -16 ± 207 | -617 ± 78 | 160 ± 72 | -37 ± 151 |
| MCD | LR+D | 8 | 64 + 0 | - | ✓ | ✓ | 256 | -17 ± 201 | -616 ± 75 | 161 ± 71 | -40 ± 144 |
| MCD | LR+D | 8 | 64 + 0 | - | ✓ | ✗ | 512 | -39 ± 179 | -626 ± 68 | 158 ± 70 | -63 ± 113 |
| MCD | LR+D | 8 | 64 + 0 | - | ✓ | ✓ | 512 | -39 ± 179 | -626 ± 68 | 158 ± 70 | -63 ± 113 |
| MCD | LR+D | 8 | 64 + 0 | - | ✗ | - | 576 | -40 ± 177 | -627 ± 67 | 158 ± 70 | -65 ± 110 |
| MCD | LR+D | 8 | 72 + 0 | - | ✗ | - | 648 | -45 ± 174 | -629 ± 66 | 158 ± 70 | -71 ± 102 |
| MCD | LR+D | 8 | 80 + 0 | - | ✗ | - | 720 | -48 ± 171 | -631 ± 64 | 157 ± 69 | -75 ± 96 |
| MCD | LR+D | 8 | 88 + 0 | - | ✗ | - | 792 | -51 ± 168 | -632 ± 63 | 157 ± 69 | -79 ± 91 |
| MCD | LR+D | 8 | 96 + 0 | - | ✗ | - | 864 | -54 ± 166 | -633 ± 62 | 156 ± 69 | -82 ± 86 |
| MCD | LR+D | 8 | 104 + 0 | - | ✗ | - | 936 | -56 ± 164 | -634 ± 61 | 156 ± 69 | -85 ± 83 |
| MCD | LR+D | 8 | 112 + 0 | - | ✗ | - | 1008 | -58 ± 163 | -635 ± 60 | 156 ± 68 | -87 ± 80 |
| MCD | LR+D | 8 | 120 + 0 | - | ✗ | - | 1080 | -60 ± 162 | -636 ± 59 | 156 ± 68 | -89 ± 77 |
| MCD | LR+D | 8 | 128 + 0 | - | ✗ | - | 1152 | -61 ± 160 | -637 ± 59 | 155 ± 68 | -90 ± 75 |
| MCD | LR+D | 8 | 64 + 1 | $\mathbb{E}[W]$ | ✗ | - | 72 | 120 ± 415 | -565 ± 203 | 170 ± 100 | 176 ± 528 |
| MCD | LR+D | 12 | 64 + 0 | - | ✓ | ✗ | 8 | 308 ± 596 | -475 ± 296 | 184 ± 93 | 931 ± 1753 |
| MCD | LR+D | 12 | 64 + 0 | - | ✓ | ✓ | 8 | 233 ± 93 | -366 ± 55 | 206 ± 68 | 26 ± 64 |
| MCD | LR+D | 12 | 64 + 0 | - | ✓ | ✗ | 12 | 252 ± 530 | -522 ± 255 | 178 ± 89 | 681 ± 1314 |
| MCD | LR+D | 12 | 64 + 0 | - | ✓ | ✓ | 12 | 151 ± 236 | -504 ± 58 | 194 ± 71 | 17 ± 71 |
| MCD | LR+D | 12 | 64 + 0 | - | ✓ | ✗ | 16 | 216 ± 498 | -533 ± 243 | 174 ± 86 | 559 ± 1123 |
| MCD | LR+D | 12 | 64 + 0 | - | ✓ | ✓ | 16 | 135 ± 236 | -526 ± 76 | 190 ± 70 | 11 ± 76 |
| MCD | LR+D | 12 | 64 + 0 | - | ✓ | ✗ | 24 | 176 ± 463 | -550 ± 219 | 170 ± 83 | 426 ± 899 |
| MCD | LR+D | 12 | 64 + 0 | - | ✓ | ✓ | 24 | 114 ± 232 | -543 ± 84 | 185 ± 69 | 2 ± 81 |
| MCD | LR+D | 12 | 64 + 0 | - | ✓ | ✗ | 32 | 148 ± 432 | -563 ± 204 | 168 ± 81 | 345 ± 755 |
| MCD | LR+D | 12 | 64 + 0 | - | ✓ | ✓ | 32 | 98 ± 229 | -555 ± 87 | 182 ± 68 | -3 ± 88 |
| MCD | LR+D | 12 | 64 + 0 | - | ✓ | ✗ | 48 | 110 ± 385 | -579 ± 179 | 165 ± 79 | 258 ± 608 |
| MCD | LR+D | 12 | 64 + 0 | - | ✓ | ✓ | 48 | 75 ± 226 | -571 ± 89 | 178 ± 68 | -4 ± 111 |
| MCD | LR+D | 12 | 64 + 0 | - | ✓ | ✗ | 64 | 84 ± 354 | -590 ± 134 | 163 ± 77 | 203 ± 523 |
| MCD | LR+D | 12 | 64 + 0 | - | ✓ | ✓ | 64 | 58 ± 224 | -582 ± 81 | 174 ± 68 | 9 ± 153 |
| MCD | LR+D | 12 | 64 + 0 | - | ✓ | ✗ | 96 | 48 ± 302 | -604 ± 99 | 160 ± 75 | 89 ± 362 |
| MCD | LR+D | 12 | 64 + 0 | - | ✓ | ✓ | 96 | 33 ± 219 | -597 ± 71 | 169 ± 68 | 23 ± 217 |
| MCD | LR+D | 12 | 64 + 0 | - | ✓ | ✗ | 128 | 23 ± 273 | -613 ± 84 | 158 ± 74 | 28 ± 266 |
| MCD | LR+D | 12 | 64 + 0 | - | ✓ | ✓ | 128 | 14 ± 215 | -607 ± 67 | 166 ± 69 | 1 ± 202 |
| MCD | LR+D | 12 | 64 + 0 | - | ✓ | ✗ | 192 | -7 ± 239 | -624 ± 69 | 156 ± 73 | -25 ± 182 |
| MCD | LR+D | 12 | 64 + 0 | - | ✓ | ✓ | 192 | -11 ± 208 | -620 ± 62 | 161 ± 69 | -33 ± 161 |
| MCD | LR+D | 12 | 64 + 0 | - | ✓ | ✗ | 256 | -27 ± 217 | -631 ± 63 | 155 ± 72 | -47 ± 148 |
| MCD | LR+D | 12 | 64 + 0 | - | ✓ | ✓ | 256 | -28 ± 200 | -629 ± 59 | 158 ± 70 | -50 ± 139 |
| MCD | LR+D | 12 | 64 + 0 | - | ✓ | ✗ | 384 | -49 ± 188 | -639 ± 56 | 153 ± 71 | -66 ± 116 |
| MCD | LR+D | 12 | 64 + 0 | - | ✓ | ✓ | 384 | -49 ± 184 | -638 ± 55 | 154 ± 70 | -67 ± 113 |
| MCD | LR+D | 12 | 64 + 0 | - | ✓ | ✗ | 512 | -60 ± 174 | -644 ± 53 | 152 ± 71 | -76 ± 101 |
| MCD | LR+D | 12 | 64 + 0 | - | ✓ | ✓ | 512 | -60 ± 173 | -644 ± 52 | 153 ± 70 | -76 ± 100 |

Table 11 – *Continued on next page*

Table 11 – *Continued from previous page*

| Epis | Param | $R^W$ | $T$ | $P^a$ | TSVD | $\hat{D}$ | $R$ | MNIST Inpainting ×10 | CelebA Inpainting ×100 | CelebA Colorization ×1000 | Flying Chairs Optical Flow ×1000 |
|---|---|---|---|---|---|---|---|---|---|---|---|
| MCD | LR+D | 12 | 64 + 0 | - | ✓ | ✗ | 768 | -65 ± 165 | -649 ± 49 | 152 ± 70 | -82 ± 89 |
| MCD | LR+D | 12 | 64 + 0 | - | ✓ | ✓ | 768 | -65 ± 165 | -649 ± 49 | 152 ± 70 | -82 ± 89 |
| MCD | LR+D | 16 | 64 + 0 | - | ✓ | ✗ | 16 | 207 ± 452 | -529 ± 267 | 172 ± 98 | 742 ± 1397 |
| MCD | LR+D | 16 | 64 + 0 | - | ✓ | ✓ | 16 | 129 ± 155 | -535 ± 72 | 192 ± 71 | 15 ± 89 |
| MCD | LR+D | 16 | 64 + 0 | - | ✓ | ✗ | 32 | 151 ± 365 | -575 ± 168 | 163 ± 92 | 490 ± 1031 |
| MCD | LR+D | 16 | 64 + 0 | - | ✓ | ✓ | 32 | 102 ± 167 | -568 ± 76 | 182 ± 70 | 8 ± 117 |
| MCD | LR+D | 16 | 64 + 0 | - | ✓ | ✗ | 64 | 91 ± 303 | -603 ± 118 | 157 ± 87 | 304 ± 720 |
| MCD | LR+D | 16 | 64 + 0 | - | ✓ | ✓ | 64 | 65 ± 177 | -594 ± 72 | 172 ± 71 | 35 ± 212 |
| MCD | LR+D | 16 | 64 + 0 | - | ✓ | ✗ | 128 | 31 ± 249 | -624 ± 88 | 152 ± 82 | 90 ± 406 |
| MCD | LR+D | 16 | 64 + 0 | - | ✓ | ✓ | 128 | 21 ± 184 | -617 ± 68 | 162 ± 72 | 42 ± 297 |
| MCD | LR+D | 16 | 64 + 0 | - | ✓ | ✗ | 256 | -22 ± 201 | -641 ± 68 | 147 ± 78 | -13 ± 226 |
| MCD | LR+D | 16 | 64 + 0 | - | ✓ | ✓ | 256 | -24 ± 183 | -638 ± 62 | 152 ± 73 | -20 ± 208 |
| MCD | LR+D | 16 | 64 + 0 | - | ✓ | ✗ | 512 | -58 ± 166 | -655 ± 55 | 144 ± 76 | -57 ± 148 |
| MCD | LR+D | 16 | 64 + 0 | - | ✓ | ✓ | 512 | -58 ± 164 | -654 ± 54 | 146 ± 74 | -57 ± 146 |
| MCD | LR+D | 16 | 64 + 0 | - | ✓ | ✗ | 1024 | -67 ± 152 | **-663** ± 49 | **143** ± 74 | -71 ± 121 |
| MCD | LR+D | 16 | 64 + 0 | - | ✓ | ✓ | 1024 | -67 ± 152 | -663 ± 49 | 143 ± 74 | -71 ± 121 |
| SVI | D | 0 | 64 + 0 | - | - | - | 0 | 439 ± 621 | -340 ± 254 | 227 ± 104 | 655 ± 876 |
| SVI | D | 0 | 64 + 1 | $\mathbb{E}[W]$ | - | - | 0 | 457 ± 619 | -333 ± 266 | 229 ± 105 | 667 ± 852 |
| SVI | LR+D | 8 | 8 + 0 | - | ✗ | - | 72 | 268 ± 500 | -565 ± 175 | 167 ± 102 | 407 ± 845 |
| SVI | LR+D | 8 | 16 + 0 | - | ✗ | - | 144 | 167 ± 382 | -594 ± 128 | 160 ± 95 | 175 ± 479 |
| SVI | LR+D | 8 | 24 + 0 | - | ✗ | - | 216 | 115 ± 331 | -610 ± 106 | 156 ± 91 | 85 ± 339 |
| SVI | LR+D | 8 | 32 + 0 | - | ✗ | - | 288 | 84 ± 305 | -620 ± 90 | 154 ± 89 | 40 ± 272 |
| SVI | LR+D | 8 | 40 + 0 | - | ✗ | - | 360 | 63 ± 289 | -628 ± 82 | 152 ± 87 | 11 ± 229 |
| SVI | LR+D | 8 | 48 + 0 | - | ✗ | - | 432 | 49 ± 277 | -633 ± 75 | 151 ± 86 | -8 ± 198 |
| SVI | LR+D | 8 | 56 + 0 | - | ✗ | - | 504 | 38 ± 269 | -637 ± 70 | 150 ± 85 | -22 ± 176 |
| SVI | LR+D | 8 | 64 + 0 | - | ✓ | ✗ | 8 | 472 ± 842 | -481 ± 324 | 183 ± 116 | 1389 ± 2388 |
| SVI | LR+D | 8 | 64 + 0 | - | ✓ | ✓ | 8 | 311 ± 388 | -483 ± 64 | 191 ± 101 | 336 ± 610 |
| SVI | LR+D | 8 | 64 + 0 | - | ✓ | ✗ | 16 | 424 ± 782 | -516 ± 273 | 174 ± 109 | 858 ± 1619 |
| SVI | LR+D | 8 | 64 + 0 | - | ✓ | ✓ | 16 | 301 ± 413 | -536 ± 156 | 182 ± 98 | 282 ± 566 |
| SVI | LR+D | 8 | 64 + 0 | - | ✓ | ✗ | 32 | 356 ± 707 | -553 ± 218 | 168 ± 103 | 542 ± 1103 |
| SVI | LR+D | 8 | 64 + 0 | - | ✓ | ✓ | 32 | 269 ± 417 | -560 ± 149 | 174 ± 94 | 241 ± 526 |
| SVI | LR+D | 8 | 64 + 0 | - | ✓ | ✗ | 64 | 266 ± 588 | -583 ± 157 | 162 ± 98 | 303 ± 717 |
| SVI | LR+D | 8 | 64 + 0 | - | ✓ | ✓ | 64 | 218 ± 413 | -584 ± 123 | 166 ± 91 | 195 ± 488 |
| SVI | LR+D | 8 | 64 + 0 | - | ✗ | - | 576 | 29 ± 263 | -641 ± 66 | 149 ± 84 | -31 ± 163 |
| SVI | LR+D | 8 | 72 + 0 | - | ✗ | - | 648 | 22 ± 258 | -644 ± 63 | 148 ± 84 | -39 ± 150 |
| SVI | LR+D | 8 | 80 + 0 | - | ✗ | - | 720 | 16 ± 253 | -646 ± 60 | 148 ± 83 | -45 ± 141 |
| SVI | LR+D | 8 | 88 + 0 | - | ✗ | - | 792 | 12 ± 250 | -648 ± 58 | 147 ± 83 | -50 ± 133 |
| SVI | LR+D | 8 | 96 + 0 | - | ✗ | - | 864 | 8 ± 247 | -650 ± 57 | 147 ± 83 | -55 ± 126 |
| SVI | LR+D | 8 | 104 + 0 | - | ✗ | - | 936 | 4 ± 245 | -652 ± 55 | 146 ± 82 | -58 ± 121 |
| SVI | LR+D | 8 | 112 + 0 | - | ✗ | - | 1008 | 2 ± 242 | -653 ± 54 | 146 ± 82 | -61 ± 116 |
| SVI | LR+D | 8 | 120 + 0 | - | ✗ | - | 1080 | -1 ± 241 | -654 ± 53 | 146 ± 82 | -64 ± 112 |
| SVI | LR+D | 8 | 64 + 1 | $\mathbb{E}[W]$ | ✗ | - | 72 | 243 ± 599 | -558 ± 222 | 164 ± 103 | 346 ± 854 |
| DE | D | 0 | 64 + 0 | - | - | - | 0 | 249 ± 403 | -374 ± 159 | 213 ± 72 | 332 ± 492 |
| DE | LR+D | 8 | 8 + 0 | - | ✗ | - | 72 | 124 ± 352 | -578 ± 130 | 171 ± 86 | 201 ± 516 |
| DE | LR+D | 8 | 16 + 0 | - | ✗ | - | 144 | 54 ± 266 | -597 ± 103 | 166 ± 78 | 65 ± 298 |
| DE | LR+D | 8 | 24 + 0 | - | ✗ | - | 216 | 17 ± 231 | -608 ± 88 | 163 ± 75 | 11 ± 220 |
| DE | LR+D | 8 | 32 + 0 | - | ✗ | - | 288 | -13 ± 205 | -617 ± 75 | 160 ± 72 | -28 ± 167 |
| DE | LR+D | 8 | 40 + 0 | - | ✗ | - | 360 | -29 ± 193 | -623 ± 68 | 159 ± 71 | -46 ± 141 |
| DE | LR+D | 8 | 48 + 0 | - | ✗ | - | 432 | -40 ± 185 | -627 ± 65 | 158 ± 70 | -58 ± 124 |
| DE | LR+D | 8 | 56 + 0 | - | ✗ | - | 504 | -49 ± 177 | -629 ± 61 | 157 ± 69 | -68 ± 109 |
| DE | LR+D | 8 | 64 + 0 | - | ✓ | ✗ | 8 | 244 ± 519 | -529 ± 224 | 178 ± 83 | 740 ± 1454 |
| DE | LR+D | 8 | 64 + 0 | - | ✓ | ✓ | 8 | 182 ± 115 | -479 ± 70 | 205 ± 62 | 24 ± 109 |
| DE | LR+D | 8 | 64 + 0 | - | ✓ | ✗ | 16 | 198 ± 461 | -554 ± 190 | 173 ± 80 | 438 ± 912 |
| DE | LR+D | 8 | 64 + 0 | - | ✓ | ✓ | 16 | 128 ± 191 | -538 ± 92 | 190 ± 66 | 55 ± 211 |
| DE | LR+D | 8 | 64 + 0 | - | ✓ | ✗ | 32 | 134 ± 384 | -576 ± 148 | 168 ± 76 | 211 ± 552 |
| DE | LR+D | 8 | 64 + 0 | - | ✓ | ✓ | 32 | 97 ± 230 | -568 ± 99 | 180 ± 67 | 50 ± 238 |
| DE | LR+D | 8 | 64 + 0 | - | ✓ | ✗ | 64 | 67 ± 308 | -596 ± 101 | 164 ± 73 | 71 ± 332 |
| DE | LR+D | 8 | 64 + 0 | - | ✓ | ✓ | 64 | 51 ± 221 | -590 ± 82 | 172 ± 67 | 13 ± 210 |
| DE | LR+D | 8 | 64 + 0 | - | ✗ | - | 576 | -54 ± 170 | -631 ± 60 | 157 ± 68 | -76 ± 97 |
| DE | LR+D | 8 | 72 + 0 | - | ✗ | - | 648 | -58 ± 165 | -633 ± 58 | 157 ± 68 | -81 ± 91 |
| DE | LR+D | 8 | 80 + 0 | - | ✗ | - | 720 | -62 ± 162 | -634 ± 57 | 157 ± 67 | -85 ± 84 |
| DE | LR+D | 8 | 88 + 0 | - | ✗ | - | 792 | -66 ± 158 | -635 ± 55 | 157 ± 67 | -89 ± 77 |
| DE | LR+D | 8 | 96 + 0 | - | ✗ | - | 864 | -70 ± 156 | -636 ± 54 | 157 ± 67 | -92 ± 73 |
| DE | LR+D | 8 | 104 + 0 | - | ✗ | - | 936 | -72 ± 155 | -637 ± 53 | 157 ± 67 | -94 ± 69 |
| DE | LR+D | 8 | 112 + 0 | - | ✗ | - | 1008 | -75 ± 148 | -637 ± 52 | 157 ± 66 | -97 ± 65 |
| DE | LR+D | 8 | 120 + 0 | - | ✗ | - | 1080 | **-77** ± 145 | -638 ± 52 | 157 ± 66 | **-99** ± 63 |

Table 11: Extended Ablation. We compare non-Bayesian networks with aleatoric uncertainty only and various Bayesian networks with both kinds of uncertainties and various hyperparameters.

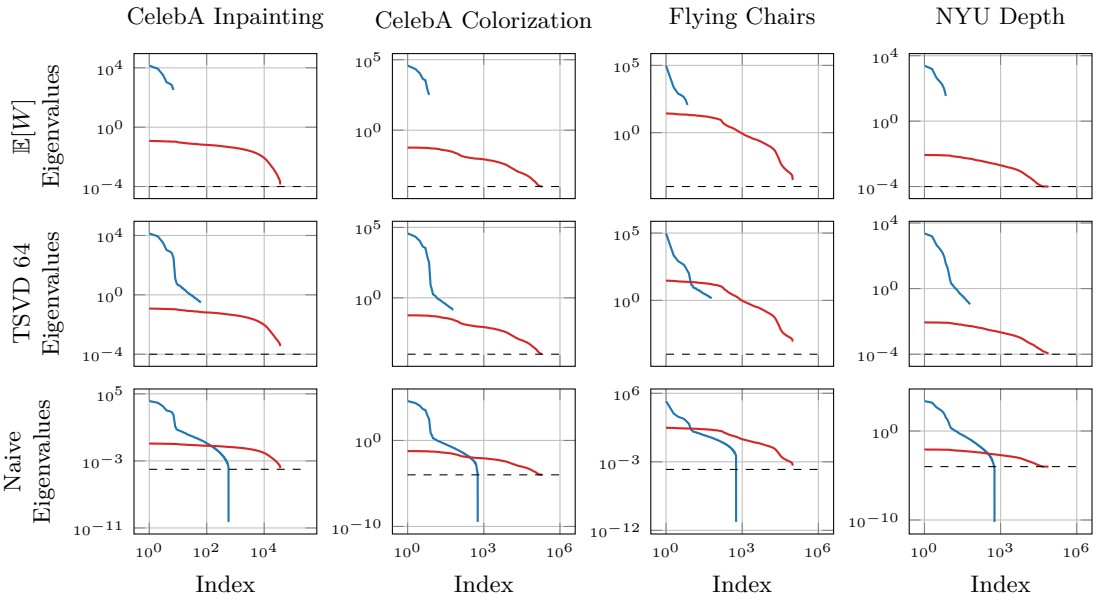

Figure 12: Normalized eigenvalues of low-rank and diagonal matrices across datasets and approximation methods.

### D.5 Eigenvalues Distribution

The figure shows the normalized eigenvalues of the low-rank matrix $PP^\top$ and the diagonal matrix $D$ (colored lines), along with a minimum diagonal threshold (dashed line), for four datasets (columns) and three different approximation methods (rows). Both axes—eigenvalue magnitude and eigenvalue index—are displayed on a logarithmic scale, with eigenvalues sorted from largest to smallest.

The top row corresponds to the expected weights $\mathbb{E}[w]$ approximation (similar to Zepf et al. (2024)), the middle row shows truncated SVD, and the bottom row displays the naive approach without truncation. The diagonal elements for the naive and expected weights rows are identical, whereas in the truncated SVD row, the diagonal is updated post-truncation.

The number of eigenvalues of the low-rank matrix depends on the rank (number of columns in $P$) and thus varies across approximation methods. The minimum diagonal entry acts as a lower bound for the diagonals in the non-updated rows (top and bottom), and the smallest eigenvalues approach this minimum.

## E  Stability

### E.1  Numerical Validation of Covariance Condition Number Bounds

To validate the reliability of our condition number estimates and stability bounds, we utilize an inpainting task on the MNIST dataset as a toy example. [2] In this setup, we train a reconstruction model to inpaint distorted handwritten digits where 5/7 of the image area is masked. We use the standard test set and a 50,000/10,000 split for training and validation, respectively. This low-dimensional setting ($S = 580$) allows us to compute the exact condition numbers of the full covariance matrix $\Sigma$ to verify our derived bounds.

Figure 14 (left) illustrates condition numbers of the covariance matrix $\kappa(\Sigma)$ for MNIST inpainting. The red and green curves represent the upper and lower bounds on the condition number, respectively, while the black

---

[2]The visual artifacts observed near the boundaries in the uncertainty maps for MNIST are primarily boundary effects resulting from the specific architecture used for this small-scale dataset. To maintain the original output resolution with a smaller model, a higher ratio of spatial padding was required during the upsampling layers compared to the larger architectures used for the higher-dimensional datasets.

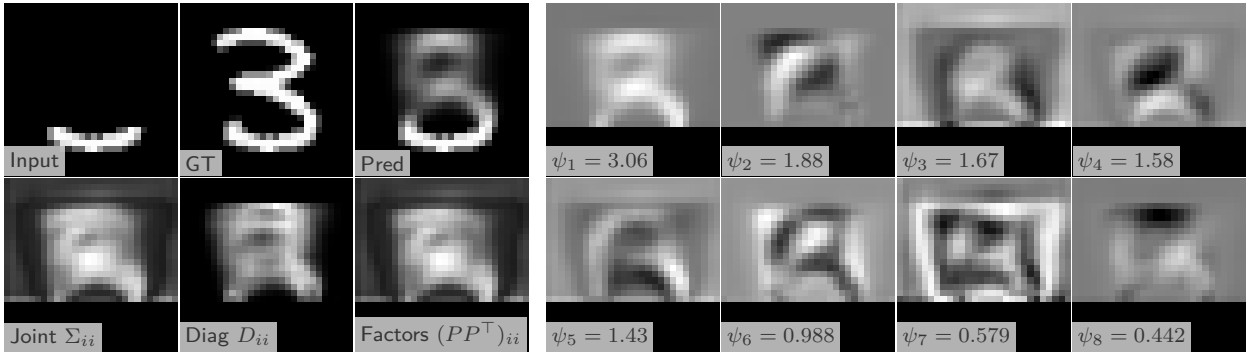

Figure 13: Qualitative Results: MNIST Inpainting. This toy example illustrates the predictive distribution for a single test digit. The first three columns show the masked input, ground truth, and predictive mean (Pred). The bottom row visualizes the marginal variances $(\Sigma_{ii}, D_{ii}, (PP^\top)_{ii})$. The subsequent columns display the 8 most significant eigenvectors of the joint low-rank matrix $P$ in descending order. Even in this simplified setting, the first eigenvectors capture structural global correlations (e.g., the overall shape and stroke continuity of the digit), while later factors represent localized variations.

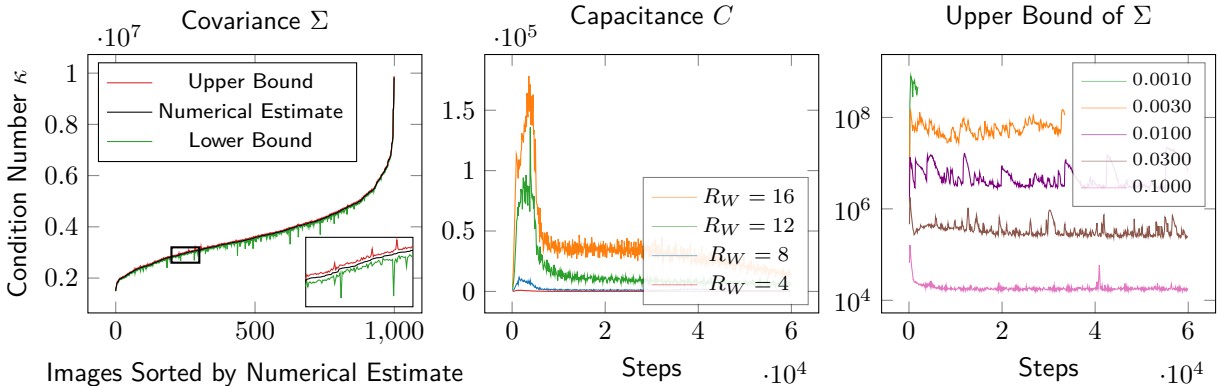

Figure 14: Condition number distributions for $\Sigma$ in MNIST inpainting (left), and effects of model parameters on condition numbers and training stability in CelebA colorization (center and right). The left panel displays condition numbers for samples ordered by their upper bound, showing exact values tightly between the computed upper and lower bounds, which validates the numerical estimates. The center panel plots the numerical condition number estimate of the capacitance matrix over training steps for varying ranks of the low-rank matrix $P_W$; higher ranks increase condition numbers and cause training instability. The right panel shows the upper bound on the covariance matrix condition number versus the minimum diagonal entry $\epsilon$ over training steps; smaller $\epsilon$ worsens conditioning and raises numerical failure risk.

curve shows the numerical estimate computed in double precision (float64) for 1000 images, sorted by their estimated condition number. Despite the inherent rounding issues in finite-precision arithmetic, the exact condition number remains tightly enclosed between the bounds, as seen particularly in the magnified section.

## E.2 Effect of Training Hyperparameters on Conditioning and Stability

Training low-rank plus diagonal (LR+D) parameterized distributions can be numerically unstable for certain hyperparameter settings. Instabilities arise during Cholesky decomposition and inversion of the capacitance matrix for log-likelihood computation.

Figure 14 (center and right) illustrates how two key parameters affect training stability. The center plot shows the estimated condition number of the capacitance matrix over time for the CelebA colorization task. Increasing the number of columns in the low-rank matrix $P_W$ raises this condition number. Models with 20 or more columns consistently crash because of exploding gradients early in training across datasets. Notably, the condition number peaks early in training and decreases after 10,000 steps, suggesting early instability.

The right plot shows the upper bound of the condition number of the full covariance matrix in MNIST inpainting when varying the minimum diagonal value added to ensure positive definiteness. Smaller values increase the condition number and often cause training failures—values below 0.001 consistently led to crashes across five random seeds.

These observations highlight a key trade-off: while small diagonal offsets enable the model to express high certainty, they also increase numerical instability. To prevent this, it is crucial to enforce a small positive $\epsilon$ on the diagonal, ensuring the smallest eigenvalue remains bounded away from zero.

### E.3 Practical Guidelines for Practitioners

To address the numerical instabilities inherent in high-dimensional joint uncertainty estimation—specifically NaN gradients for the covariance factor P and Cholesky errors during log-likelihood evaluation—we recommend the following staged guideline and stabilization techniques.

**Phased Model-by-Model Training** Rather than training the full joint Bayesian model from scratch, stability is best achieved through a sequence of discrete model runs. This allows the practitioner to identify the stability boundary for their specific domain:

1. **Deterministic Diagonal Baseline.** Train a non-Bayesian model predicting only the mean $\mu$ and diagonal uncertainty $D(x)$.

2. **Deterministic Low-Rank (LR+D) Integration.** Transition to an LR+D model incorporating the factor $P(x)$. Start with conservative hyperparameters: Rank $R \in [4, 10]$, $\epsilon = 0.01$, and a low loss weight $\alpha = 1/16$. This prioritizes mean accuracy while introducing dominant correlations in a numerically safe environment.

3. **Joint Bayesian Framework.** Once the deterministic LR+D structure is stable, enable Bayesian components (e.g., MC Dropout or Ensembles). This stage may require its own refinement: consider a lower dropout rate or lower prior standard deviation initially to maintain stability.

4. **Iterative Refinement.** In subsequent models, incrementally increase $\alpha$ or R and decrease $\epsilon$.

   - **Monitoring Rank Value:** Measure if higher R reduces the NLL. Tasks that are less multi-modal often perform optimally with a low R.
   - **Monitoring Eigenvalues:** Track how many diagonal entries in $D(x)$ are determined by the $\epsilon$ bound. If many "hit the floor," $\epsilon$ can be lowered; if the model becomes unstable, the precision limit of the architecture has been reached.

**Robustness Tricks** We suggest several practical safeguards to ensure higher training robustness:

- **Gradient Filtering:** Implement a mechanism to catch numerical exceptions. The gradients for the covariance factor P are particularly prone to NaNs. If an error is detected for a specific batch, skip the update step for those occurrences to prevent model collapse. This is particularly logical if only a few steps throughout the entire training process result in NaNs.

- **Full SVD on P during Training:** To avoid Cholesky errors, apply a full SVD on the factor P without dimensionality reduction during the forward pass. Since P is not sampled during training (unlike the final joint covariance), the dimensionality is usually feasible. While this slows down training, it significantly improves numerical conditioning.

**Stability in Inference and Sampling**    The choice of the number of posterior samples T during inference also impacts stability:

- **With SVD:** Using TSVD during the sampling process ensures high stability even for large T, though it carries a higher computational cost.

- **Without SVD:** If SVD is not used, a lower T is more likely to remain numerically stable than a high T, as it reduces the probability of encountering an ill-conditioned sample in the posterior distribution.

## F    Algorithm

The algorithm constructs a low-rank plus diagonal covariance matrix using Monte Carlo sampling from a proxy posterior distribution over network weights. It begins by performing $T$ forward passes, each with weights sampled from the proxy distribution $q_\theta^*$. For each pass, the predictive mean $\mu_{w_i}(x)$, aleatoric uncertainty factor $P_{w_i}^a(x)$, and diagonal covariance $D_{w_i}(x)$ are computed. These estimates are then aggregated by averaging over samples to obtain $\mu(x)$ and $D(x)$.

Next, the epistemic uncertainty factor $P^e(x)$ is constructed by stacking the centered deviations of the predictive means from their average, scaled appropriately. The aleatoric factor $P^a(x)$ is either averaged or stacked similarly to capture inherent noise uncertainty. The total low-rank factor matrix $P(x)$ is formed by concatenating these epistemic and aleatoric components.

Optionally, truncated SVD can be applied to $P(x)$ to reduce its rank and improve computational efficiency. The final covariance estimate $\Sigma(x)$ is assembled as the sum of the low-rank product $P(x)P(x)^\top$ and the diagonal $D(x)$. This low-rank decomposition enables scalable and expressive uncertainty quantification by efficiently combining both epistemic and aleatoric sources.

## G    Derivations in Detail

### G.1    Exploiting LR+D for efficient computation of matrix determinant and inverse

Both the likelihood function $p(y|x,w) = \mathcal{N}\left(\mu_W^a(x), \Sigma_W^a(x)\right)$ as well as the approximate posterior predictive distribution $p(y|x,\mathcal{D}) \approx \mathcal{N}\left(\mu(x), \Sigma(x)\right)$ are multivariate normal distributions parametrized by covariance matrices $\Sigma_W^a$ and $\Sigma$, respectively, where in the following, we only consider $\Sigma$ for clarity. Denoting by $S$ the output dimension, the normal distribution is then defined as

$$\mathcal{N}\left(\mu(x), \Sigma(x)\right) = \frac{1}{\sqrt{|\Sigma(x)|(2\pi)^S}} \exp\left(-\frac{1}{2}(\mu(x) - y)^\top \Sigma^{-1}(x)(\mu(x) - y)\right) \tag{16}$$

which requires computation of the covariance matrix' determinant $|\Sigma|$ and inverse $\Sigma^{-1}$ for sampling and evaluation of the log-likelihood. For full covariance matrices $\Sigma \in \mathbb{R}^{S \times S}$ with large $S$, these are very expensive, if not impossible, to compute directly. Instead, we exploit our LR+D representation for efficient computation of the matrix determinant and inverse.

We compute the determinant as

$$|\Sigma| = |D + PP^\top| \tag{17}$$
$$= |I_R + P^\top D^{-1}P||D| \tag{18}$$
$$= |C||D| \tag{19}$$

where we first substituted $\Sigma$ with its LR+D representation and subsequently applied the matrix determinant lemma. With $D \in \mathbb{R}^{S \times S}$, $P \in \mathbb{R}^{S \times R}$ and $I_R \in \mathbb{R}^R$, the so-called capacitance $C = I_R + P^\top D^{-1}P$ is an $R \times R$ matrix. Since $R \ll S$, the determinant of the capacitance matrix is very cheap to compute.

---

**Algorithm 2** LR+D Covariance Construction with Monte Carlo Sampling

---

**Require:** Proxy posterior $q_\theta^*$
**Require:** Input $x$
**Ensure:** Mean $\mu(x)$, diagonal $D(x)$, low rank factor $P(x)$, covariance $\Sigma(x)$
    // **Step 1: Monte Carlo Sampling**
1: **for** $i = 1$ **to** $T$ **do**
2:     Sample weights $w_i \sim q_\theta^*$
3:     Compute $\mu_{w_i}(x)$, $P_{w_i}(x)$, $D_{w_i}(x)$
4: **end for**
    // **Step 2: Aggregate Means**
5: $\mu(x) = \frac{1}{T} \sum_{i=1}^{T} \mu_{w_i}(x)$
6: $D(x) = \frac{1}{T} \sum_{i=1}^{T} D_{w_i}(x)$
    // **Step 3: Epistemic Factor**
7: $P^e(x) = \frac{1}{\sqrt{T-1}} \begin{bmatrix} \mu_{w_1}(x) - \mu(x) & \mu_{w_2}(x) - \mu(x) & ... & \mu_{w_T}(x) - \mu(x) \end{bmatrix}$
    // **Step 4: Aleatoric Factor**
8: $P^a(x) = \frac{1}{T} \begin{bmatrix} P_{w_1}(x) & P_{w_2}(x) & ... & P_{w_T}(x) \end{bmatrix}$
    // **Step 5: Combine Factors**
9: $P(x) \leftarrow \begin{bmatrix} P^a(x) & P^e(x) \end{bmatrix}$
    // **Step 6: Optional SVD**
10: **if** use SVD **then**
11:     $[U, S, \_] = \text{svd}(P(x))$
12:     $\hat{P}(x) \leftarrow U_{:,1:r} S_{1:r,1:r}$
13:     $\hat{D}_{ii}(x) \leftarrow D_{ii}(x) + \sum_{j=r+1}^{R} U_{ij}^2 S_{jj}^2$
14: **end if**
    // **Final Covariance**
15: $\Sigma(x) = P(x)P(x)^\top + D(x)$

---

To compute the inverse, we use the Woodbury matrix identity, again by exploiting the LR+D representation.

$$\Sigma^{-1} = (D + PP^\top)^{-1} \tag{20}$$
$$= D^{-1} - D^{-1}P(I_R + P^\top D^{-1}P)^{-1}P^\top D^{-1} \tag{21}$$
$$= D^{-1} - D^{-1}PC^{-1}P^\top D^{-1} \tag{22}$$

As before, the capacitance matrix $C \in \mathbb{R}^{R \times R}$ is very small and thus its inverse easy to compute.

### G.2 Full derivation of SVD

We apply dimensionality reduction using SVD on our tall $P$ matrices. This involves decomposing into three separate matrices: $U$, $\Psi$, and $V^\top$. The $U$ matrix represents an arbitrary not further used rotation, $\Psi$ is a diagonal matrix containing the singular values, and $V^\top$ contains the columns of the transformed matrix.

By selecting the top $R$ singular values and corresponding vectors, we can approximate the original matrix. This approximation is achieved by truncating the matrices $U$ and $V^\top$ to retain only the top $R$ singular values and vectors. This reduces the dimensionality of the data while preserving its essential structure.

The reduced dimensionality representation, denoted as $\hat{P}$, is computed by taking the product of the truncated matrices $V$ and $\Psi$. Additionally, a diagonal matrix $\hat{D}$ captures the by the dimensionality reduction removed variance of $PP^\top$, with each element representing the contribution of the omitted singular values to the overall

uncertainty. We use $\hat{D}$ to update our diagonal for the final LR+D representation.

$$P^\top = U\Psi V^\top \tag{23}$$

$$PP^\top = (U\Psi V^\top)^\top (U\Psi V^\top) \tag{24}$$

$$= V\Psi U^\top U\Psi V^\top \tag{25}$$

$$= V\Psi\Psi V^\top \tag{26}$$

$$\hat{\Sigma} = \hat{D} + \hat{P}\hat{P}^\top \tag{27}$$

$$\hat{P} = \begin{bmatrix} V_{R-\hat{R}} \cdot \Psi_{R-\hat{R},R-\hat{R}} & \cdots & V_R \cdot \Psi_{R,R} \end{bmatrix} \tag{28}$$

$$\hat{D}_{ii} = \sum_{j=1}^{R-\hat{R}-1} V_{ij}^2 \cdot \Psi_{j,j}^2 \tag{29}$$

### G.3    Loss Definition

For regression problems we intend to maximize the data likelihood $p(\boldsymbol{y}|\boldsymbol{x}, w) = \prod_i p(y_i|x_i, w)$, where we assumed all dataset samples to be i.i.d. Equivalently, we can minimize the negative log-likelihood $p(\boldsymbol{y}|\boldsymbol{x}, w) = \sum_i -\log p(y_i|x_i, w)$. We further assume the network predictions to be distributed around the true value $y$ following a Gaussian distribution with mean $\mu_w(x)$ and covariance $\Sigma_w(x)$.

The subscript $()_w$ implies given weights, as in a classical frequentist interpretation of neural networks. To account for the epistemic part of uncertainty, this weight is not fixed, and a probabilistic interpretation is used, implying a $p(w)$. Later, we approximate the posterior of $p(w|\mathcal{D})$ by using Monte Carlo integration. If we drop $\sigma_w$ as nuisance parameter, we would recover a standard squared error loss as it is often used in regression.

The loss function for a single training sample can then be defined as

$$\begin{aligned}
\mathcal{L}_{lrd} &= -\frac{1}{S}\log p(y|x,w) \\
&= -\frac{1}{S}\log\left(\frac{1}{\sqrt{|\Sigma_w|(2\pi)^S}}\exp\left(-\frac{1}{2}(\mu_w - y)^\top \Sigma_w^{-1}(\mu_w - y)\right)\right) \\
&= \frac{1}{S}\left(\log\sqrt{|\Sigma_w|(2\pi)^S} + \frac{1}{2}(\mu_w - y)^\top \Sigma_w^{-1}(\mu_w - y)\right) \\
&= \frac{1}{S}\left(\frac{1}{2}\log|\Sigma_w| + \frac{S}{2}\log(2\pi) + \frac{1}{2}(\mu_w - y)^\top \Sigma_w^{-1}(\mu_w - y)\right)
\end{aligned}$$

where we normalized by the output dimensionality $S$.

Dropping constant terms, we are left with:

$$\mathcal{L}_{lrd} = \frac{1}{2S}\log|\Sigma_w| + \frac{1}{2S}(\mu_w - y)^\top \Sigma_w^{-1}(\mu_w - y)$$

We can see that evaluating $\mathcal{L}$ involves computing the determinant and inverse of the covariance matrix. To achieve this, we exploit our LR+D representation as described in the previous section G.1

### G.4 Full derivation of mean vector and covariance matrix

The expectation of the posterior predictive distribution is given by:

$$\mathbb{E}[y|x, \mathcal{D}] = \mathbb{E}_{p(w|\mathcal{D})}\left[\mathbb{E}\left[y|x, w\right]\right] \tag{30}$$

$$\approx \mathbb{E}_{q_\theta^*}\left[\mathbb{E}\left[y|x, w\right]\right] \tag{31}$$

$$= \mathbb{E}_{q_\theta^*}\left[\mu_W^a\left(x\right)\right] \tag{32}$$

$$\approx \frac{1}{T}\sum_i^T \mu_{w_i}^a(x) \quad w_i \sim q_\theta^* \tag{33}$$

$$= \mu(x) \tag{34}$$

The covariance of the posterior predictive distribution is given by:

$$\text{Cov}\left[y|\,x, \mathcal{D}\right] = \text{Cov}_{p(w|\mathcal{D})}\left[\mathbb{E}_{p(w|\mathcal{D})}\left[y|x, w\right]\right] + \mathbb{E}_{p(w|\mathcal{D})}\left[\text{Cov}\left[y|x, w\right]\right] \tag{35}$$

$$\approx \text{Cov}_{q_\theta^*}\left[\mathbb{E}_{q_\theta^*}\left[y|x, w\right]\right] \qquad + \mathbb{E}_{q_\theta^*}\left[\text{Cov}\left[y|x, w\right]\right] \tag{36}$$

$$= \underbrace{\text{Cov}_{q_\theta^*}\left[\mu_W^a(x)\right]}_{epistemic} \qquad + \underbrace{\mathbb{E}_{q_\theta^*}\left[\Sigma_W^a(x)\right]}_{aleatoric} \tag{37}$$

$$\approx \Sigma^e(x) \qquad\qquad + \Sigma^a(x) \tag{38}$$

$$= \Sigma(x) \tag{39}$$

$$\Sigma^e(x) = \frac{1}{T-1}\sum_i^T \left(\mu_{w_i}\left(x\right) - \mu\left(x\right)\right)\left(\mu_{w_i}\left(x\right) - \mu\left(x\right)\right)^\top \quad w_i \sim q_\theta^* \tag{40}$$

$$\Sigma^a(x) = \frac{1}{T}\sum_i^T \Sigma_{w_i}(x) \quad w_i \sim q_\theta^*. \tag{41}$$

In above transformations of expectation and variance, we applied the law of total expectation or variance, respectively, and subsequently approximated them using the proxy distribution $q_\theta^\star(w)$. The expectation over $y$, denoted by $\mathbb{E}\left[y|x, w\right]$, is given by the mean of the predicted normal distribution $\mu_W^a\left(x\right)$, whereas the covariance over $y$, denoted as $\text{Cov}\left[y|x, w\right]$, is given by the covariance matrix of the predicted normal distribution. Finally, the expectation – and in some suggested solutions also the covariance – over the proxy distribution $q_\theta^\star(w)$ is approximated using Monte Carlo integration, i.e. sampling $T$ weights from the proxy $w \sim q_\theta^\star$.

### G.5 Bounds for the Condition Number

The exact condition number of $\Sigma$ can be calculated using the following equation:

$$\kappa(\Sigma) = \frac{\lambda_S(\Sigma)}{\lambda_1(\Sigma)}$$

Therefore, we need to estimate both the smallest and largest eigenvalues of $\Sigma$.

We begin by analyzing the eigenvalues of the individual components of the LR+D decomposition. The eigenvalues of the diagonal matrix $D$ are simply its diagonal entries,

$$\lambda_i(D) = \{D_{ii}|D_{i-1,i-1} \geq D_{ii} \geq D_{i+1,i+1} > 0\}, \tag{42}$$

where the diagonal entries are assumed to be sorted in non-increasing order without loss of generality.

The symmetric matrix $PP^\top$ is rank-deficient, with at most $R$ non-zero singular values. Using the Singular Value Decomposition (SVD), its eigenvalues are given by

$$\lambda_i(PP^\top) = \begin{cases} 0, & \text{if } i \leq S - R, \\ \Psi_{jj}^2, & \text{otherwise, where } j = i + S - R \end{cases} \tag{43}$$

where the non-zero eigenvalues are sorted in non-increasing order.

Direct computation of the eigenvalues of the full covariance matrix $\Sigma = PP^\top + D$ is computationally prohibitive. Instead, we approximate or bound them using Weyl's inequality for the eigenvalues of Hermitian matrices (Horn & Johnson, 2012, Theorem 4.3.1), which provides an efficient way to estimate the eigenvalues of matrix sums. For $1 \leq i \leq S$, the eigenvalues of $\Sigma$ satisfy the following sandwich inequality:

$$\lambda_{i-j+1}(PP^\top) + \lambda_j(D) \leq \lambda_i(\Sigma) \leq \lambda_{i+k}(PP^\top) + \lambda_{S-k}(D), \tag{44}$$

where $j$ and $k$ can be chosen freely within the ranges $1 \leq j \leq i$ and $0 \leq k \leq S - i$.

To simplify the calculation, we assume that the $R$ largest eigenvalues of $PP^\top$ dominate the diagonal entries of $D$. Furthermore, the remaining eigenvalues of $PP^\top$ are zero and thus smaller than any positive eigenvalue of $D$. Based on this assumption, we propose fixed choices of $j$ and $k$ to yield the following bounds:

$$\left. \begin{array}{r} \lambda_1(D) \\ \lambda_{i-R}(D) \\ \lambda_i(PP^\top) \quad + \quad \lambda_1(D) \end{array} \right\} \leq \lambda_i(\Sigma) \leq \begin{cases} \lambda_{i+R}(D) & \text{if } i \leq R \\ \lambda_{i+R}(D) & \text{if } i \leq S - R \\ \lambda_i(PP^\top) \quad + \quad \lambda_S(D) & \text{if } S - R < i \end{cases} \tag{45}$$

Finally, the condition number $\kappa(\Sigma)$ can be bounded by substituting the eigenvalue bounds:

$$\frac{\lambda_S(PP^\top) + \lambda_1(D)}{\lambda_{R+1}(D)} \leq \kappa(\Sigma) \leq \frac{\lambda_S(PP^\top) + \lambda_S(D)}{\lambda_1(D)}. \tag{46}$$

These bounds provide a computationally efficient means of estimating the condition number without requiring exact eigenvalue decomposition, making them well-suited for large-scale covariance matrices in deep learning applications.

### G.6 Alternative Approximation of the Condition Number

In the paper, we suggest using bounds to estimate both the eigenvalues and the condition number. However, an approximate solution using power iteration is also available. To estimate the largest and smallest eigenvalues, we first need to approximate their corresponding eigenvectors $v_S$ and $v_1$. This approximation requires the covariance matrix and its inverse, respectively. One of the main advantages of the LR+D parametrization is its efficient inverse. The following equations approximate the desired eigenvectors using $i$ as iteration step.

$$v_S^{i+1} = \frac{\Sigma v_S^i}{||\Sigma v_S^i||_2} \tag{47}$$

$$v_1^{i+1} = \frac{\Sigma^{-1} v_S^i}{||\Sigma^{-1} v_S^i||_2} \tag{48}$$

After convergence, the eigenvalues can be calculated $\lambda_S = ||\Sigma v_S||$ and $\lambda_1 = ||\Sigma v_1||$. The cost of the matrix multiplication $v\Sigma$ and $v_\Sigma^{-1}$ can be reduced by the creation of intermediate arrays. In the naive form. Due to further optimization, we can reduce the memory $\mathcal{O}(S^2)$ and time $\mathcal{O}(\max(R^2, S)SI)$ complexity to $\mathcal{O}(SR)$ and $\mathcal{O}(SR \max(I, R))$, where $I$ is the number of iteration steps. After the optimization, The equation for the eigenvector corresponding to the largest eigenvalue looks like:

$$\Sigma v_i = (D + PP^\top)v_i \tag{49}$$

$$= Dv_i + P(P^\top v_i), \tag{50}$$

where the equation for the smallest eigenvalue looks like:

$$\Sigma^{-1} v_i = (D + PP^\top)^{-1} v_i \tag{51}$$

$$= (D^{-1} - (D^{-1}P(I_R + P^\top D^{-1}P)^{-1})(P^\top D^{-1}))v_i \tag{52}$$

$$= \underbrace{D^{-1}}_{\substack{\mathbb{R}^{S x S} \\ diagonal}} v_i - \underbrace{(D^{-1}P(I_R + P^\top D^{-1}P)^{-1})}_{\mathbb{R}^{S x R}} \underbrace{((P^\top D^{-1}))}_{\mathbb{R}^{R x S}} v_i). \tag{53}$$

The underbraced parts can be precomputed and reused for every iteration.

Stop the power iteration when the residual norm falls below a tolerance:

$$\|\Sigma v_S^i - \lambda_S^i v_S^i\|_2 < \epsilon_\kappa \tag{54}$$

or equivalently via the Rayleigh quotient change:

$$\left| \frac{v_S^{i\top} \Sigma v_S^i}{v_S^{i\top} v_S^i} - \frac{v_S^{i-1\top} \Sigma v_S^{i-1}}{v_S^{i-1\top} v_S^{i-1}} \right| < \epsilon_\kappa . \tag{55}$$

The same criteria apply analogously to $v_1^i$, replacing $\Sigma$ with $\Sigma^{-1}$ and $\lambda_S$ with $\lambda_1$.

The convergence is geometric and the ratio for the largest eigenvalue is given by $c = \frac{\lambda_S}{\lambda_{S-1}}$, whereas for the smallest eigenvalue it is given by $c = \frac{\lambda_2}{\lambda_1}$. A primary limitation of this approximation is that the convergence of the smallest eigenvalue is quite slow, as the smaller eigenvalues are numerically similar $\lambda_2 \sim \lambda_1$.

