# OpenReview forum: "It depends: Incorporating correlations for joint aleatoric and epistemic uncertainties of high-dimensional output spaces"
_TMLR — Accepted by TMLR_

### Review · Reviewer_iKii · 2026-02-01

**Summary Of Contributions:**

The authors propose to jointly model aleatoric and epistemic uncertainty with the aim of caturing correlations between output variables in challenging domains such as image inpainting, colorization or optical flow prediction.
The proposed method is agnositic to the actually used posterior sampling method and demonstrated with the widely used MCD, SVI and DE methods.
They both quantitatively and qualitatively analyze their method, showing that it outperforms not modeling output correlations.
Finally, they propose to use truncation and introduce stabilization tricks to scale the method.

**Strengths:**

- Agnostic to the posterior sampling method
- relatively stable w.r.t. R (Figure 6)
- Interesting choice of benchmarks, also w.r.t. the qualitative results!

**Weaknesses**

- Depth estimation benchmark as in Kendall and Gal 2017 could have been incorporated for comparison.
- Hard to judge the significance of the results as test set variability, not over multiple runs is provided for quantitative results. See below.

**Additional Comments:**

It would be interesting to see how the method performs in widely used last-layer setting, where posterior sampling and thus representing epistemic uncertainty is only done on the final layer of the network.
This setting is widely used in practice for feasibility of high fidelity posterior sampling and it would be interesting to know what benefit the method can provide there.

**Audience:**

Yes

**Audience Explanation:**

Capturing corrlations between output varianbles for uncertainty estimation in multivariate prediction settings is an often overlooked, but potentially impactful area in the years ahead. The LR+D approach presented in this paper is actionable. I also found the presented stabilization tricks valuable - estimating higher moments of predictive distributions in regression can be very challenging in practice!

**Broader Impact Concerns:**

N.A.

**Claims And Evidence:**

No

**Claims Explanation:**

The major concern I have is the use of **test set variability** instead of variability over multiple runs of the method in the quantitative results.
Other than that, I think that convincing evidence is given for the performance of the method.

**Requested Changes:**

I would love a discussion on why the artifacts at the bottom left and right for the diagonal term appear for the MNIST examples (e.g. Figure 3 MNIST Diag or Figure 10 for all Diag outputs).

I found Figure 1 not very intuitive, maybe a better description in the caption helps, or a rework of the example. Especially make clear what a sample means here.

Both are not critical for whether or not the work should be accepted, but would further strengthen its contribution.

---

> ### Author Response · Authors · 2026-04-01
> **Depth Estimation Benchmark, Multi-Seed Evaluation, and Structural Revisions**
>
> We thank the reviewer for the valuable comments and ideas. In particular, we found the suggestion to use depth estimation as a potential application task very appealing.
>
> ### Benchmark and Experimental Updates
> As proposed by the reviewer, we have replaced the MNIST benchmark in the main paper with a **depth estimation benchmark** (Kendall and Gal, 2017). We believe this adds significantly more value and strengthens the paper, as the MNIST benchmark (with 28x28 pixels reduced to an even smaller inpainting task) was arguably not representative of truly high-dimensional settings.
>
> *Note: We have retained the MNIST task in **Section E of the Appendix** specifically to analyze the numerical stability (condition number) of the covariance. This remains necessary as it is the only task where we can computationally handle a dense covariance matrix for comparison due to the smaller output space.*
>
> ---
>
> ### Response to Requested Changes
>
> * **Variability Across Runs:** To address the concern regarding test set variability, we have updated our quantitative results to reflect performance over **5 different random seeds**. This captures the variability across independent runs for MC Dropout, SVI, and the diagonal baseline. For Deep Ensembles, the ensemble itself represents the integrated variability; therefore, individual seed-based reruns for the entire ensemble were omitted due to the extreme computational requirements of these high-dimensional tasks.
> * **Artifacts in MNIST:** We clarified that the artifacts observed in the marginal uncertainty maps for MNIST (e.g., Figure 3) are primarily boundary effects resulting from the specific architecture used for that dataset. To maintain the original output resolution with a smaller model, a higher ratio of spatial padding was required during the upsampling layers compared to the larger architectures used for the higher-dimensional datasets.
> * **Revision of Figure 1:** We agree with the reviewer that Figure 1 required more context. We have revised the figure along with its description to offer a more intuitive and understandable version of the proposed framework.

---

### Review · Reviewer_SPCW · 2026-02-19

**Summary Of Contributions:**

Summary:

The authors study the problem of uncertainty estimation for regression tasks with high-dimensional output spaces.

They combine sampling-based methods for epistemic uncertainty estimation (Monte Carlo dropout, ensembling, SVI) with a Gaussian likelihood, using a low-rank + diagonal covariance matrix output by the network (rather than the standard approach of using a diagonal matrix).

Their proposed method is evaluated on image inpainting (MNIST, CelebA-HQ), image colorization (CelebA-HQ) and optical flow estimation (Flying Chairs), in terms of the log-likelihood. They compare against a diagonal covariance matrix baseline.

***






Strengths:
- The paper is quite well written overall, very few typos or similar issues.

- The studied problem of uncertainty estimation for regression, and specifically for high-dimensional regression tasks such as image inpainting/colorization, is interesting.

- The proposed method is quite interesting, to use a low-rank + diagonal covariance matrix rather than just a diagonal (thus being able to capture correlations) makes sense overall.

***












Weaknesses:
- At least personally, I found it somewhat difficult to follow/understand all details of the proposed method in Section 2.

- The proposed method adds substantial complexity compared to the diagonal covariance matrix baseline, involves multiple hyperparameters, and seems to be quite prone to numerical issues / training instability.

- In Section 4, the authors write that their method "_exhibits a lower negative log-likelihood and **produces more reliable uncertainty estimates**, demonstrating clear advantages in uncertainty modeling_", but I don't think this actually has been demonstrated. The methods are only evaluated in terms of log-likelihood, practical utility of the uncertainty estimates is not demonstrated in any concrete practical use case (e.g., selective prediction).

- At least personally, I find it difficult to interpret the qualitative results in Figure 3 and Figure 7 - 11. It is not clear to me how these visualizations actually should/could be used in practice.

***










Questions/suggestions:
- I think it could make more sense to move the "Hyperparameters for Stable Training" paragraph to the Methods section?

- What value do you use for $\alpha$ in the loss?

- The proposed method seems quite significantly more complex than the diagonal covariance matrix baseline, is this actually worth it in practice? Nice that it improves the test log-likelihood, but are there concrete practical use cases where your method adds significant benefits?

- How would you summarize your network output into a single, image-level uncertainty estimate? Or, a single pixel-level uncertainty estimate? This is straightforward for a diagonal covariance matrix.

- At least to me, Section 2.5 seems a bit out-of-place when reading the paper for the first time (when I got to 2.5, it wasn't really clear to me what this would be used for, or why this was included here).

***











Minor things:
- Section 1, "Yet, certain of these approaches can only model uncertainty in the local neighborhood using a band Cholesky parametrization": "certain of these" --> "some of these"?
- Section 2, "p(D|w) refers to the likelihood, that represents the epistemic part of the uncertainty": Shouldn't this be aleatoric?
- Section 2.4, "The full matrix P (x), representing the joint aleatoric uncertainty" --> "The full matrix P (x), representing the joint uncertainty"?
- Section 4, "Kendall & Gal (2017); Monteiro et al. (2020); Duff et al. (2023)": Incorrect citation formatting.

**Audience:**

Yes

**Audience Explanation:**

The studied problem of uncertainty estimation for regression, and specifically for high-dimensional regression tasks such as image inpainting/colorization, is interesting.

**Broader Impact Concerns:**

No concerns.

**Claims And Evidence:**

No

**Claims Explanation:**

In Section 4, the authors write that their method "_exhibits a lower negative log-likelihood and **produces more reliable uncertainty estimates**, demonstrating clear advantages in uncertainty modeling_", but I don't think this actually has been demonstrated. The methods are only evaluated in terms of log-likelihood, practical utility of the uncertainty estimates is not demonstrated in any concrete practical use case (e.g., selective prediction).

**Requested Changes:**

This is a quite well-written paper overall that I think could be relevant for the TMLR audience.

However, I think the current version requires some clarifications and modifications, see "Weaknesses" and "Questions/suggestions" above.

---

> ### Author Response · Authors · 2026-04-01
> **Improved Intuition, Practical Utility, and Metric Clarifications**
>
> We thank the reviewer for the valid comments and suggestions and aim to address them in the revised version.
>
> ### Improved Intuition and Exposition
> The reviewer pointed out that it may be difficult to follow/understand the proposed method. We have added further clarifications at the beginning of the **Methods section** regarding the core concept. Furthermore, we updated **Figure 1** along with its description to provide better intuition for the framework.
>
> ### Addressing Complexity and Practical Utility
> The reviewer noted that the method adds substantial complexity compared to simpler methods. We now explicitly acknowledge this in the **Discussion**. To assist users, we added **Appendix E.3**, which provides concrete, recipe-style guidelines on how to implement the method and handle the associated challenges. We also added additional descriptions and discussion regarding the results produced by the proposed method.
>
> ---
>
> ### Questions and Suggestions (Q&A)
>
> * **Move "Hyperparameters for Stable Training" to Methods:** We agree this improves the flow. We have revised the structure so the **Methods section** now contains a dedicated part discussing numerical stability, including the hyperparameters for stable training.
> * **Value of $\alpha$:** We trained the models using a value of $\alpha = 0.125$. This information has been added to the paragraph on model specifics in **Section 3**.
> * **Practical Use Cases:** We have expanded the discussion on the practical utility of our method in two key areas:
>     1.  **Downstream Aggregation and Systemic Risk:** We updated the **Discussion (Section 4)** to highlight the necessity of joint uncertainty. We specifically address how our method prevents the underestimation of uncertainty during spatial aggregation—a critical requirement for medical diagnostics and infrastructure monitoring where pixel covariance significantly impacts the reliability of the aggregate sum.
>     2.  **Selective Prediction Performance:** In **Appendix D.2**, we provide an empirical evaluation of our uncertainty estimates for the "reject option." We demonstrate via Coverage–Risk curves that our differential entropy is a robust predictor of the Negative Log-Likelihood, allowing the model to effectively identify and abstain from low-confidence predictions.
> * **Summarizing Joint Uncertainty:** Communicating uncertainty for high-dimensional tasks is challenging. In **Appendix D.2**, we discuss using **differential entropy** as a summary statistic. Using the **Matrix Determinant Lemma**, we compute this entropy efficiently for LR+D matrices. This provides a more reliable signal for model "abstention"—identifying when a prediction is likely to be coherently wrong across an entire region.
> * **Pixel-level Uncertainty Estimate:** While our method captures joint correlations, it is straightforward to extract the marginal pixel-wise uncertainty by taking the diagonal of the joint covariance matrix $\Sigma = PP^T + D$. We have added these visualizations to the qualitative results in **Figure 3** to provide a familiar point-wise uncertainty map. However, we emphasize that this diagonal view alone neglects the spatial correlations which are the core of our method.
> * **Section 2.5 Placement:** We have added more context to this section to highlight its significance and broader scope, aligning it with the methodological contributions as part of the primary Methods discussion.
> * **Minor Corrections:** We thank the reviewer for noticing these details; we have made the suggested changes accordingly.

---

> > ### Comment · Reviewer_SPCW · 2026-04-05
> >
> > Thank you for the response.
> >
> > I have read the other reviews and all rebuttals.
> >
> > The other reviewers are quite positive overall, and I think the authors provided a comprehensive and solid rebuttal.
> >
> > I will recommend accept.

---

### Review · Reviewer_t4Su · 2026-02-19

**Summary Of Contributions:**

The paper proposes a framework for jointly modeling aleatoric and epistemic uncertainty using low-rank plus diagonal (LR+D) approximations of output covariance matrices. Leveraging the law of total variance, the authors decompose the predictive covariance into epistemic and aleatoric components, applying LR+D parameterizations to each. To reduce the computational burden associated with high sample counts in Bayesian deep learning methods such as MC Dropout, the framework further employs a truncated singular value decomposition. In the evaluation, the authors consider multiple Bayesian deep learning techniques, i.e., MC Dropout, deep ensembles, and SVI, and assess their approach on three datasets spanning different tasks (inpainting, colorization, and optical flow estimation). Quantitative results demonstrate that incorporating epistemic uncertainty yields higher log-likelihoods on unseen test samples. Additional experiments illustrate the practical benefits of the proposed approximation in terms of time and space complexity.

**Audience:**

Yes

**Audience Explanation:**

The paper addresses a practically relevant problem at the intersection of uncertainty quantification and structured prediction, and I believe it is well-suited for the TMLR audience.

**Claims And Evidence:**

No

**Claims Explanation:**

**Strengths:**

- The paper is well-written and easy to follow.
- The framework is broadly applicable: it can be combined with any Bayesian inference method, provided it is computationally feasible for the model at hand. In particular, widely used approaches such as MC Dropout are readily supported.
- The unified, end-to-end formulation that jointly predicts the mean and covariance is appealing, as it provides a principled and coherent approach to modeling both aleatoric and epistemic uncertainty.
- Code is publicly available, supporting reproducibility.
- The evaluation is comprehensive, spanning three datasets with distinct tasks (inpainting, colorization, and optical flow estimation).

**Weaknesses:**
- The paper does not clearly explain how the proposed framework improves upon or differs from existing work that also employs LR+D covariance approximations. The authors themselves acknowledge in the related work section that several prior approaches (Salinas et al., 2019; Monteiro et al., 2020; Willette et al., 2021; Stoica & Babu, 2023) already use low-rank plus diagonal parameterizations to capture global correlations. Given this, the method section should explicitly highligh what is novel about the present contribution relative to these works. For instance, whether the novelty lies in the decomposition into epistemic and aleatoric components, the truncated SVD for computational savings. Without such a discussion, it is difficult for the reader to assess the true contribution and novelty of the work.
- Given the acknowledged numerical instabilities, the paper would benefit from a concrete recommendation paragraph on training and hyperparameter selection.  For practitioners applying this framework to new domains, it remains unclear how to choose key hyperparameters, such as the number of posterior samples $T$ and the rank $R$, as well as appropriate training settings. A practical guideline or a more in-depth sensitivity analysis to out-of-distribution data would substantially improve the framework's accessibility beyond the tasks considered here.
- The contribution of Section 2.5 remains unclear. The section serves more as a diagnostic tool and does not explain the concrete steps taken to address numerical instabilities. The authors state that these bounds "can be used to monitor and mitigate numerical instabilities," yet the section reads more like a description of how instability could be detected than of how it is prevented in practice.

**Minor Issues:**
- The introduction claims that diagonal covariance assumptions lead to "miscalibrated uncertainties and incoherent predictions in structured settings," yet the evaluation does not include calibration metrics to substantiate this.
- The paper frequently refers to "high-dimensional outputs" without specifying at what dimensionality the diagonal assumption becomes problematic. Concrete thresholds or examples would strengthen the argument.
- On page 3, the authors state that the likelihood represents the epistemic component of uncertainty, which appears to be a typo.
- Page 3, Section 2, second line: "denoting" should read "denotes."
- What is the design rationale for not including a diagonal covariance term for the epistemic uncertainty component?
- On page 11, $T$ is frequently not typeset in math mode.

Overall, I consider this a solid and well-presented paper. I am willing to accept the paper if the weaknesses outlined above are adequately addressed in the rebuttal.

**Requested Changes:**

See Weaknesses.

---

> ### Author Response · Authors · 2026-04-01
> **Clarifications on Novelty, Stability, and Practical Guidelines**
>
> We thank the reviewer for their constructive comments. They were very helpful in strengthening the paper in terms of communication and understandability. We address the mentioned weaknesses and issues as follows:
>
> ---
>
> ### Weakness 1: Communication objections with respect to other methods and concrete contributions
>
> The idea to use a low-rank plus diagonal (LR+D) form is indeed well-established and widely used to represent covariance matrices. The key differences are that:
>
> * **Comprehensive Uncertainty Representation:** We present a method that represents both epistemic **and** aleatoric uncertainty for problems with very large covariance matrices, capturing full uncertainty rather than only components. This is achieved by finding a solution to propagate epistemic variation into the LR+D form along with the aleatoric part.
> * **Truncated SVD Approximation:** We utilize a truncated SVD approach as a soft approximation to handle the high dimensionality of the output space. Unlike methods that neglect epistemic variation or rely on a simple $\mathbf{E}[W]$ approximation (expected value of weights), we recognize that since aleatoric uncertainty is itself a model prediction, there is an inherent epistemic uncertainty associated with that prediction. The SVD allows us to capture the most significant components of this variation across posterior samples and integrate them into the LR+D form, ensuring the method remains computationally tractable and numerically stable even with tens of thousands of output dimensions.
>
> We updated the manuscript in the **Contribution** paragraph (Section 1), the **Related Work** section (Section 1), and the opening of **Section 2** to communicate this more directly.
>
> ---
>
> ### Weakness 2: Practical guidelines for practitioners
>
> We added **Section E.3** in the appendix to provide concrete, recipe-style guidelines to offer better accessibility for practitioners.
>
> ---
>
> ### Weakness 3: Unclear contribution of Section 2.5
>
> We address this in two ways:
> 1. We updated **Section 2.5** to better communicate the context and align it with the rest of the paper, highlighting its role as a methodological contribution.
> 2. We refer to Section 2.5 in the newly introduced guideline section (Appendix E.3) to clarify its practical application in a more concise way.
>
> ---
>
> ### Minor Issues
>
> * **Evaluation Metrics:** We prioritize Test Log-Likelihood (TLL) as it is a "proper scoring rule" for multivariate Gaussian distributions, directly accounting for the correlations between pixels which is the core of our contribution. In contrast, standard metrics like Expected Calibration Error (ECE) focus on pixel-wise marginals and ignore the covariance structure. We evaluate the multivariate TLL for both diagonal and non-diagonal covariance matrices to ensure a fair comparison.
> * **Definition of "high-dimensional outputs":** We added concrete thresholds and information in the **Introduction** and the **Discussion**.
> * **Typo for likelihood:** We corrected the typo as suggested.
> * **General Typos:** We have addressed the noted typos throughout the text.
> * **Absence of Diagonal in the epistemic part:** The epistemic component of our covariance is derived from sample-based variations and is inherently rank-deficient, as the number of posterior samples ($T$) is typically much smaller than the output dimension ($D$). Incorporating a separate diagonal term for the epistemic part is dispensable for the likelihood calculation, as the aleatoric diagonal $D$ already provides the necessary regularization to ensure full rank. Furthermore, obtaining a reliable epistemic diagonal would be computationally prohibitive, requiring a significantly larger number of samples to avoid noisy estimates.
> * **Notation:** We corrected the math mode for $T$ as requested.

---

### Decision · Action_Editor_SHyZ · 2026-04-16

**Recommendation:** Accept as is

**Audience:**

Yes

**Audience Explanation:**

The paper addresses uncertainty quantification in high-dimensional regression, a topic of clear relevance to the TMLR audience.

**Claims And Evidence:**

Yes

**Claims Explanation:**

The revised manuscript addresses the main concern regarding evaluation soundness by reporting variability across multiple runs, and the evidence is now sufficient to support the claims.